# Unraveling a bifunctional mechanism for methanol-to-formate electro-oxidation on nickel-based hydroxides

Botao Zhu[1], Bo Dong[1], Feng Wang[1], Qifeng Yang[1], Yunpeng He[1], Cunjin Zhang[2], Peng Jin [2] ✉ & Lai Feng [1] ✉

For nickel-based catalysts, in-situ formed nickel oxyhydroxide has been generally believed as the origin for anodic biomass electro-oxidations. However, rationally understanding the catalytic mechanism still remains challenging. In this work, we demonstrate that NiMn hydroxide as the anodic catalyst can enable methanol-to-formate electro-oxidation reaction (MOR) with a low cell-potential of 1.33/1.41 V at 10/100 mA cm$^{-2}$, a Faradaic efficiency of nearly 100% and good durability in alkaline media, remarkably outperforming NiFe hydroxide. Based on a combined experimental and computational study, we propose a cyclic pathway that consists of reversible redox transitions of Ni$^{II}$-(OH)$_2$/Ni$^{III}$-OOH and a concomitant MOR. More importantly, it is proved that the Ni$^{III}$-OOH provides combined active sites including Ni$^{III}$ and nearby electrophilic oxygen species, which work in a cooperative manner to promote either spontaneous or non-spontaneous MOR process. Such a bifunctional mechanism can well account for not only the highly selective formate formation but also the transient presence of Ni$^{III}$-OOH. The different catalytic activities of NiMn and NiFe hydroxides can be attributed to their different oxidation behaviors. Thus, our work provides a clear and rational understanding of the overall MOR mechanism on nickel-based hydroxides, which is beneficial for advanced catalyst design.

Hydrogen (H$_2$) as a carbon-free and high-energy density fuel has been considered one of the most important alternatives to conventional fossil fuels[1]. Recently, the overall water splitting (OWS) technique has been well developed[2,3], which enables the green and sustainable generation of H$_2$ and benefits instituting a zero-carbon society. Nevertheless, the OWS normally requires a cell-potential higher than 1.6 V due to the sluggish oxygen evolution reaction (OER) with a high standard potential of 1.23 V$_{SHE}$ at the anodic side[4,5]. Systematic cost analysis reveals that the energy consumed for OER occupies ca. 95% of total energy for OWS[6], leading to low cost-efficiency of H$_2$ generation. To further increase the cost-efficiency, it is highly desired to replace energy-consuming OER with other anodic reactions with low energy consumption and/or with a value-added product.

Methanol (350 $ per ton) has been long used as a precursor to produce formate (1300 $ per ton) with higher value, an important chemical in rubber and pharmaceutical industries[7,8]. However, conventional synthesis of formate involves multiple steps under harsh conditions[9]. Alternatively, methanol-to-formate electro-oxidation reaction (MOR) has been of great interest very recently due to the mild synthesis condition and low standard potential of 0.103 V$_{SHE}$[5,10,11]. A variety of electro-catalysts have been designed as the anode to boost MOR and hence to realize the combined production of H$_2$ and value-added product of formate[12–15].

[1]Soochow Institute for Energy and Materials Innovation (SIEMIS), School of Energy, Soochow University, Suzhou, China. [2]School of Materials Science and Engineering, Hebei University of Technology, Tianjin, China. ✉e-mail: china.peng.jin@gmail.com; fenglai@suda.edu.cn

Nickel-based anode catalysts have been widely used for OER and various biomass electro-oxidations. As many biomass (i.e., methanol, ethanol, furfural, and furfuryl amine) have a nucleophilic group (i.e., hydroxyl, aldehyde, and amino groups), their oxidations are defined as nucleophile oxidation reactions (NOR)[16–20]. To more efficiently produce value-added products from biomass, many efforts have been devoted to understanding the NOR process[21–25]. In a state-of-the-art-work, Wang et al. focused on the change of $Ni(OH)_2$ catalyst under the EOR conditions and proposed a one-electron reaction consisting of an endothermic oxidation of $Ni^{II}$-$(OH)_2$ to $Ni^{III}$-OOH and an exothermic or spontaneous reduction of $Ni^{III}$-OOH to $Ni^{II}$-$(OH)_2$ along with the dehydrogenation of ethanol[23]. Subsequently, Shalom et al. surveyed on the electrocatalytic process of MOR and revealed that the overall rate of MOR is bottlenecked by the C−H bond oxidation of methanol, rather than the formation of $Ni^{III}$-OOH on the surface of $NiFeO_x$[24]. Another group also afforded similar results based on an uphill energy barrier upon the conversion from *$OCH_2$ to *OCH during the MOR of $Ni(OH)_2$ with $Ni^{III}$-OOH species[25]. Thus, the in-situ formed $Ni^{III}$-OOH was generally believed as the origin of MOR or EOR. However, experimental identification was blocked due to the transient stability of $Ni^{III}$-OOH, especially under optimal MOR or EOR conditions (i.e., at a potential of <1.5 $V_{RHE}$ with a high alcohol concentration). To date, how these transient species promote alcohol oxidation remains ambiguous. The most typical mechanism proposed the electrophilic oxygen of $Ni^{III}$-OOH as the single active site towards MOR[25], which may well account for the selective formation of formate but not the rapidly diminishing of $Ni^{III}$-OOH during the MOR. To further address both phenomena, an alternative catalytic mechanism is required.

In this work, we employ NiM-LDHs (M = Mn, Fe, LDH: layered double hydroxide) as model catalysts toward MOR. As compared to NiFe-LDH, NiMn-LDH exhibits enhanced MOR activity, requiring only 1.33/1.41 V to reach 10/100 mA cm$^{−2}$ with a formate Faradaic efficiency of nearly 100% and good durability in alkaline media. Operando Raman spectroscopic observations reveal, for the first time, the transient formation of $Ni^{III}$-OOH on the surface of NiM-LDHs under optimal MOR conditions. Meanwhile, a H/D kinetic isotope effect (KIE) study is employed to identify the potential-determining step (PDS) and rate-determining step (RDS). Furthermore, based on the experimental results, density functional theory (DFT) computations suggest a cyclic pathway consisting of reversible $Ni^{II}/Ni^{III}$ redox transitions and a concomitant MOR. More importantly, it is verified that the electro-catalytic MOR involves two active sites including $Ni^{III}$ and nearby electrophilic oxygen species of $Ni^{III}$-OOH, which work in a cooperative manner to promote the MOR. Such a mechanism is different from that based on a single active site and hence defined as the bifunctional mechanism by considering the combined functionalities of both active sites[26].

## Results

### Synthesis and characterizations

A series of NiMn and NiFe-LDHs have been synthesized on the surface of nickel foam (NF) using a modified hydrothermal method. The scanning electron microscopy (SEM) images (Fig. 1a, b and Supplementary Fig. 1) show that both LDHs are formed as nanosheets, which densely and uniformly cover the surface of NF. Their nanosheet-like morphologies were confirmed by transmission electron microscopy (TEM) images (Supplementary Fig. 2). The high-resolution TEM (HR-TEM) characterizations (Fig. 1c, e) reveal their crystalline natures with lattice spacing distances of 2.35 Å for NiMn-LDH and 2.69 Å for NiFe-LDH, in line with X-ray diffraction (XRD) analysis (Supplementary Fig. 3). The high-angle annular dark field (HAADF) image and corresponding energy dispersive spectroscopy (EDS) mapping demonstrates the uniform distribution of O, Ni, and Mn or Fe elements in NiMn or NiFe-LDH nanosheet (Fig. 1d, f). Chemical compositions of the as-synthesized NiM-LDHs were probed by using the X-ray photoelectron spectroscopy (XPS) technique, revealing the valence states of $Ni^{2+}$, $Mn^{3+}$, and $Fe^{3+}$ (see Supplementary Fig. 4, 5 and Note 1, 2). This result is in line with literature reports[27,28]. The atomic ratio of Ni/Mn and Ni/Fe were identified to be 4.4 and 4.6, respectively, by using the inductively coupled plasma-optical emission spectrometer (ICP-OES) method (Supplementary Table 1).

### Electro-catalytic performance evaluation

The MOR performance of NiMn and NiFe-LDHs were investigated in 1 M KOH solution with 3 M $CH_3OH$ using the linear sweep voltammetry (LSV) technique. NiMn-LDH exhibits a remarkable lower MOR onset potential (at 2 mA cm$^{−2}$) of 1.30 $V_{RHE}$ (or a smaller onset overpotential of 1.20 V) as compared to that (1.37 $V_{RHE}$ or 1.27 V) of NiFe-LDH, indicating that NiMn-LDH can more efficiently boost MOR under the applied conditions (Fig. 2a, Supplementary Fig. 6 and Note 3).

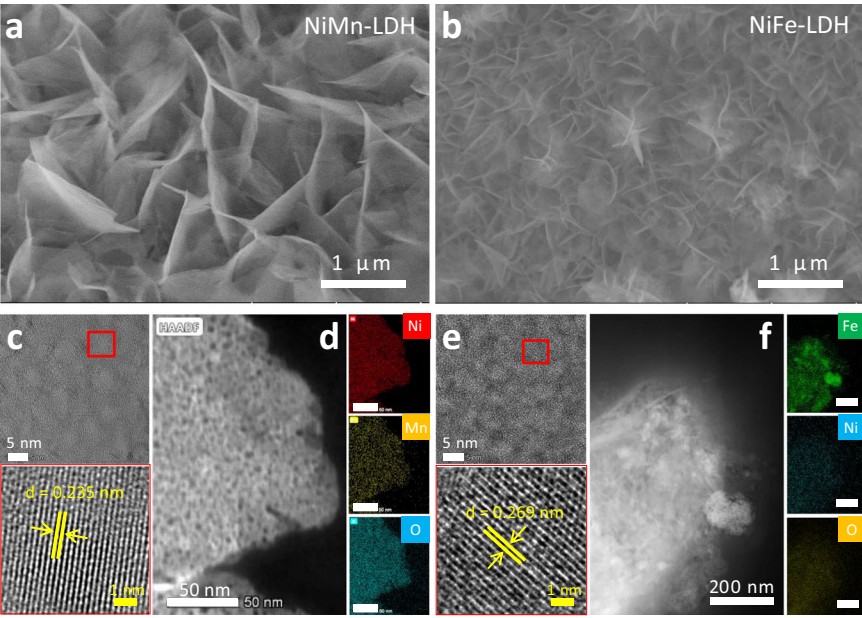

**Fig. 1 | Preparation and characterizations of the catalysts.** SEM images of the **a** NiMn and **b** NiFe-LDHs. HR-TEM and HAADF, elemental mapping images of **c**, **d** NiMn and **e**, **f** NiFe-LDHs.

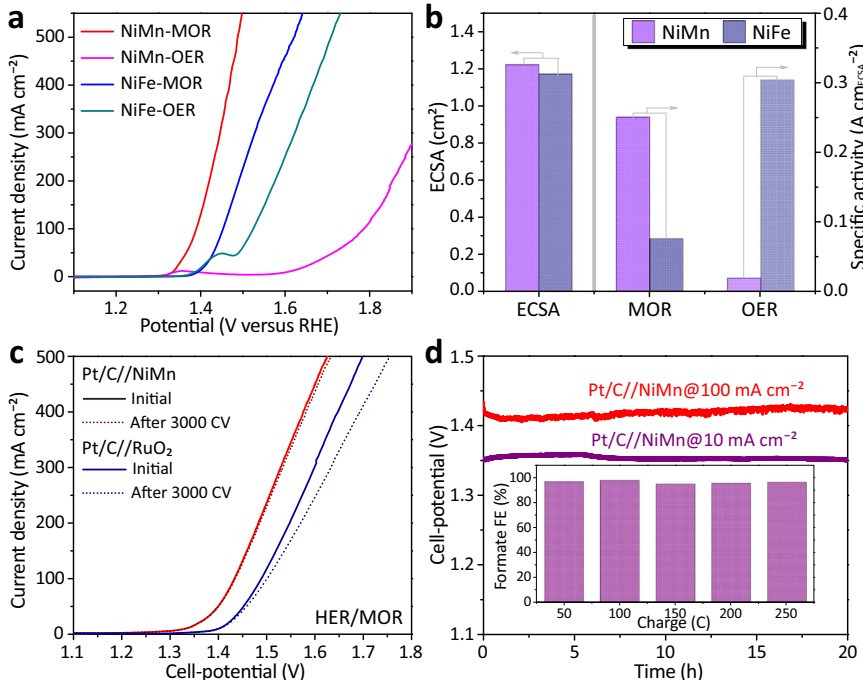

**Fig. 2 | Electrocatalytic performance of the catalysts. a** IR-corrected LSV curves of NiMn and NiFe-LDHs both recorded in 1 M KOH without and with 3 M CH₃OH. **b** ECSA normalized specific activities of NiMn and NiFe-LDHs for MOR (at 1.45 $V_{RHE}$) and OER (at 1.65 $V_{RHE}$). **c** Cell LSV curves (no iR correction) of HER/MOR electrolysis using an electrode pair of Pt/C//NiMn before and after 3000 CV sweeps in a single-chamber electrolyzer with 1 M KOH and 3 M CH₃OH, using Pt/C//RuO₂ as reference. **d** Chronopotentiometry (CP) profiles at 10/100 mA cm⁻² (inset shows FE of formate obtained via continuous electrolysis).

A similar trend is also observed at the higher current density of 100/500 mA cm⁻², where NiMn-LDH requires a lower working potential of 1.41/1.49 $V_{RHE}$ as compared to that (1.45/1.62 $V_{RHE}$) of NiFe-LDH. To rule out the impacts of LDH nanostructures and estimate the intrinsic activities of NiMn and NiFe towards MOR, their current densities are normalized by the electrochemical active surface area (ECSA) (1.22 and 1.17 cm² for NiMn and NiFe-LDH, respectively). Their ECSA are estimated by using an equation of ECSA = $C_{dl}/C_s$, where $C_{dl}$ and $C_s$ are double-layer capacitance (see Supplementary Fig. 7 and Note 4) and specific capacitance (see Supplementary Fig. 8), respectively. As plotted in Fig. 2b, NiMn delivers a specific activity of 250.1 mA cm$_{ECSA}^{-2}$ towards MOR (at 1.45 $V_{RHE}$), almost 3.3-fold higher than that (75.8 mA cm$_{ECSA}^{-2}$) of NiFe. Nevertheless, we found that NiMn-LDH is more inert than NiFe-LDH under OER conditions (Fig. 2a), in good agreement with literature reports[29,30]. By using the same method, the OER-specific activity of NiMn-LDH is almost one-sixteenth that of NiFe-LDH (Fig. 2b). Thus, the above results demonstrate that NiMn-LDH is intrinsically more active than NiFe-LDH under the MOR conditions, though it is less active for OER. These results indicate that the MOR mechanism significantly differs from that of OER.

Furthermore, the two-electrode electrolyzer for HER/MOR electrolysis was set up by using Pt/C as the cathode and NiMn-LDH as the anode to concurrently produce H₂ and formate. The polarization curves recorded before and after 3000 CV sweeps in alkaline media are provided in Fig. 2c. It is seen that the Pt/C//NiMn electrode pair required a cell-potential of 1.33/1.43 V to achieve the current density of 10/100 mA cm⁻², much lower than that (1.40/1.49 V) of a fully commercial pair of Pt/C//RuO₂. After 3000 CV sweeps, the Pt/C//NiMn pair displayed an almost constant polarization curve, indicative of its good stability in alkaline media. In comparison, Pt/C//RuO₂ pair showed remarkable decay under the same condition probably due to the gradual dissolution of Ru⁴⁺ in alkaline media[31]. The good stability of Pt/C//NiMn pair was confirmed by the chronopotentiometry (CP) tests (Fig. 2d), in which no significant increase

in cell potential was observed at 10 and 100 mA cm⁻² after 20-h electrolysis. It is clearly seen that gas bubbles (H₂) were continuously produced on the cathode, while no bubble was produced on the anode. The anodically produced formate was quantified by analyzing electrolyte using ion chromatography (IC) (Supplementary Fig. 9, 10) and its Faradaic efficiency (FE) was found to be constant with a value of nearly 100% along the continuous electrolysis (Supplementary Table 2). The electrolysis conducted in a double-chamber electrolyzer yielded similar results (Supplementary Fig. 11 and Table 3). The ¹H and ¹³C nuclear magnetic resonance (NMR) spectra (Supplementary Fig. 11d) of the electrolyte show only formate signals. As no CO₃²⁻ signal (δ = 162 ppm) could be detected, the production of CO₂ could be ruled out, again confirming the selective conversion from methanol to formate. The anode catalyst after the 20-h CP test was checked again by using SEM and XPS (Supplementary Fig. 12, 13), which showed no obvious decay in the catalyst morphology and compositions. These results confirm the long-term durability of NiMn catalyst under the MOR conditions.

## Insight into the MOR mechanism

To investigate the intermediates or structural change of NiM-LDH-based anode, we conducted a series of operando Raman measurements under the MOR conditions and compared them to those of OER. As shown in Fig. 3a and Supplementary Fig. 14, Note 5, the Raman bands of NiMn-LDH at open circuit potential (OCP) appear at 468, 533, and 600 cm⁻¹, corresponding to Niᴵᴵ-O and Mnᴵᴵᴵ-O bending and/or stretching vibrations, respectively[23,32,33]. Under the OER conditions, a pair of feature bands of Niᴵᴵᴵ-O (at 473 and 551 cm⁻¹) newly emerges at the applied potential of 1.37 $V_{RHE}$ and remains dominant in the following anodic and cathodic potential sweep (i.e., 1.42–1.52–1.12 $V_{RHE}$). This result is in line with the anode color change from brownish-yellow to black (see Supplementary Movie 1, 2), indicating that the in-situ formed Niᴵᴵᴵ-OOH could remain stable once it formed and act as the origin of OER[29]. In comparison, under the MOR conditions, the feature bands of Niᴵᴵᴵ-O also emerge at 1.37 $V_{RHE}$, which are less

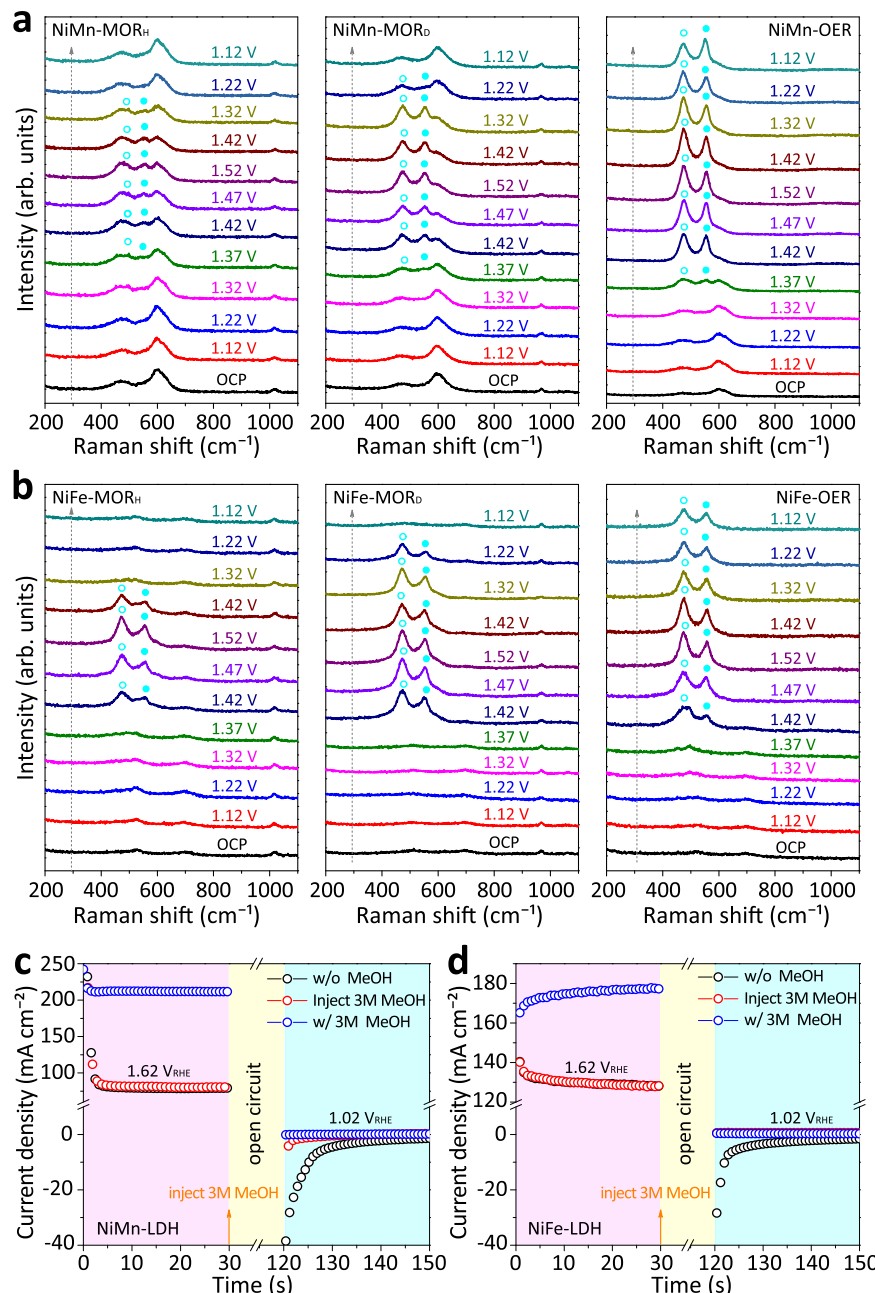

**Fig. 3 | Evidence for in-situ formation and reduction of $Ni^{III}$-OOH under the optimal MOR conditions.** Operando Raman spectra of **a** NiMn and **b** NiFe-LDHs both obtained at various potentials under optimal MOR (the subscript H or D denotes $CH_3OH/H_2O$ or $CD_3OD/D_2O$ solution) and OER conditions, respectively. The blue circles indicate the feature bands of $Ni^{III}$-OOH. Multi-potential step curves of **c** NiMn and **d** NiFe-LDHs in 1 M KOH solutions without and with 3 M $CH_3OH$. In all panels, the pink, yellowish, and pale green shaded areas indicate the application of a constant voltage of 1.62 $V_{RHE}$, an open-circuit process and the application of an open circuit voltage of 1.02 $V_{RHE}$, respectively.

remarkable relative to the primary bands of $Ni^{II}$-O in the potential range of 1.37–1.52 $V_{RHE}$. When the applied potential decreases from 1.52 to 1.12 $V_{RHE}$, the feature bands of $Ni^{III}$-O gradually weaken and vanish. All the above observations reveal that the Ni ions of NiMn-LDH shuttle between $Ni^{II}$ and $Ni^{III}$ during the MOR catalysis. Meanwhile, the $Mn^{III}$-O band (at 600 cm$^{-1}$) remains the same throughout the potential sweeping, indicating that the Mn ions of NiMn-LDH are not involved in the redox transition. To confirm the $Ni^{II}$-$(OH)_2/Ni^{III}$-OOH transition, additional operando Raman measurements were performed in deuterium media (i.e., $CD_3OD/D_2O$) under the MOR conditions. As shown in Fig. 3a, more pronounced $Ni^{III}$-O bands (at 473 and 551 cm$^{-1}$) were detected in the potential range of 1.42–1.52–1.32 $V_{RHE}$ with deuterium media, indicating that the methanol-induced reduction of $Ni^{III}$-OOH

was retarded due to the more robust O–D or C–D bond of the absorbate. Meanwhile, the color change of NiMn anode could be detected from brownish-yellow to dark and then to the original, which is otherwise negligible in the aqueous media. Additionally, $Ni_{0.85}Mn_{0.15}$-LDH with more Ni content shows more evident Raman bands of $Ni^{III}$-O under the same MOR conditions (see Supplementary Fig. 15 and Note 6), as compared to those of NiMn-LDH, which again confirms the transient formation of $Ni^{III}$-OOH. When switching the anode catalyst to NiFe-LDH, the operando Raman measurements deliver similar results (Fig. 3b, Supplementary Fig. 16 and Note 7). Thus, by using the operando spectroscopic technique, we can verify the transient and limited formation of $Ni^{III}$-OOH in NiM-LDHs under optimal MOR conditions, which is likely the origin of MOR catalysis.

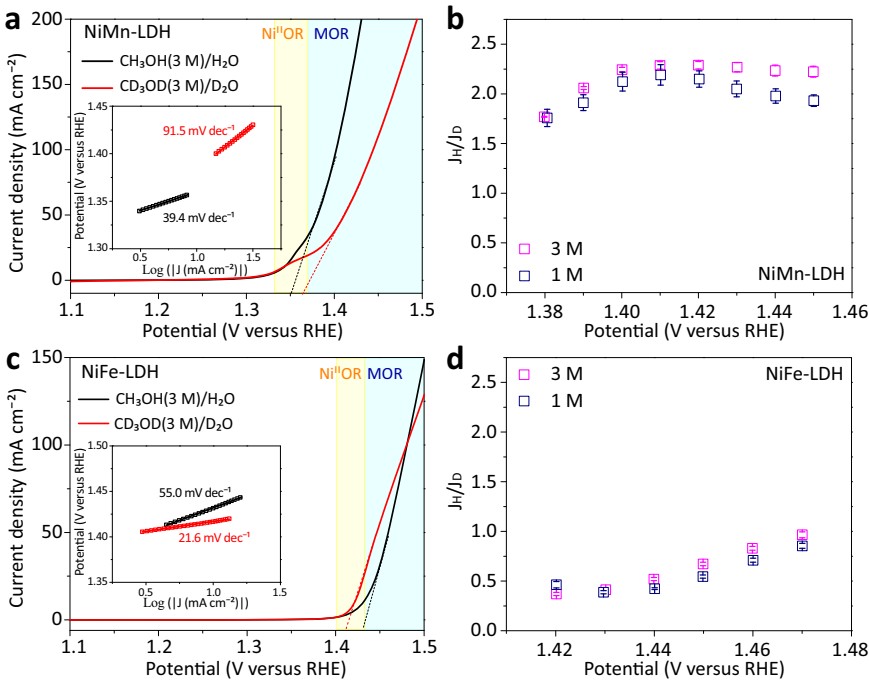

**Fig. 4 | H/D KIE studies. a** CV curves (only anodic sweep is shown for clarity) of NiMn-LDH recorded in alkaline $CH_3OH$ (3 M)/$H_2O$ and $CD_3OD$ (3 M)/$D_2O$, respectively. $Ni^{II}OR$ refers to $Ni^{II}$ oxidation reaction, and the dash line indicates the occurrance of MOR. Inset shows Tafel plots. **b** H/D KIE plots of NiMn-LDH ($J_H$ and $J_D$ refer to the current densities obtained in alkaline aqueous and deuterium solutions with $CH_3OH$ and $CD_3OD$ in different concentrations (1 or 3 M)). The error bars correspond to the standard deviations of measurements over two or three independent electrodes under the same conditions. **c** CV curves and **d** H/D KIE plots of NiFe-LDHs. In all panels, the pale yellow shaded area indicates the electrooxidation of $Ni^{II}$ to $Ni^{III}$, and the pale blue shaded area indicates MOR process.

In addition, to probe if the reduction of $Ni^{III}$-OOH by methanol is exothermic or spontaneous, intermittent MOR and OER measurements were performed by applying different potentials. As shown in Fig. 3c, d, the initially applied potential was set at 1.62 $V_{RHE}$ for NiM-LDH-based anode to generate $Ni^{III}$-OOH. Then, after an open-circuit state, the applied potential was switched to 1.02 $V_{RHE}$. Under the OER condition, a remarkable reduction current could be observed at 1.02 $V_{RHE}$ for both NiMn and NiFe-LDHs, implying the non-spontaneous reduction of $Ni^{III}$-OOH in absence of methanol[19,21]. In comparison, when methanol was added upon an open-circuit state (90s), negligible reduction current could be observed for both LDHs at 1.02 $V_{RHE}$, indicative of the spontaneous reduction of $Ni^{III}$-OOH to $Ni^{II}$-$(OH)_2$ under the MOR conditions. Alternatively, when methanol was added before measurement, a much higher current density (180–210 mA cm$^{-2}$) could be observed at 1.62 $V_{RHE}$ as compared to that (80–130 mA cm$^{-2}$) attained under OER condition, indicating the occurrence of MOR. After the open-circuit state, no current density was detected at 1.02 $V_{RHE}$, thus confirming the spontaneous reduction of $Ni^{III}$-OOH to Ni-$(OH)_2$ under the MOR conditions. Nevertheless, afterward, further dehydrogenations of absorbate are necessary for the formation of formate, which remains to be explored (*vide infra*).

Furthermore, a H/D KIE study was conducted for NiM-LDHs, which helps to identify both the PDS and RDS in the catalytic MOR reaction[34,35]. The former defines the onset potential of MOR, while the latter settles the MOR current beyond onset. Figure 4a and Supplementary Fig. 17a show the CVs starting with an anodic sweep for the overall MOR of NiMn-LDH in aqueous and deuterium media. It is seen that the $Ni^{II}$-$(OH)_2$/$Ni^{III}$-OOH oxidation occurs at a low potential of around 1.35 $V_{RHE}$, closely followed by the catalytic MOR with sharply increased current. As compared with $CH_3OH$/$H_2O$, the use of $CD_3OD$/$D_2O$ leads to a positively-shifted onset potential of the catalytic MOR along with remarkably reduced current density at the given potential of 1.37–1.50 $V_{RHE}$ and a higher Tafel slope (91.5 versus 39.4 mV dec$^{-1}$). The positively shifted onset indicates that the proton-coupled electron

transfers (PCETs: i.e., O-H(D) or C-H(D) bond breaking) are involved in the PDS of catalytic MOR[34,35]. We also quantify the KIE as $J_H/J_D > 1.5$ in the Tafel range (Fig. 4b)[36,37], regardless of the methanol concentration (Supplementary Fig. 18a and Note 8), suggesting a normal KIE. This indicates that the PCETs are also involved in the RDS of catalytic MOR, probably as a consequence of PDS[38].

On the other hand, the CVs for the overall MOR on NiFe-LDH are provided in Fig. 4c and Supplementary Fig. 17b, 18b. We can observe a very close overlap between $Ni^{II}$-$(OH)_2$/$Ni^{III}$-OOH oxidation current and catalytic MOR current, indicating the concurrent occurrence of precatalytic oxidation and catalytic MOR. Nevertheless, it is found that the onset potential of MOR is negatively shifted in the duterio media, indicating that the PCETs in the catalytic MOR are not involved in the PDS. In addition, by comparing the MOR currents and Tafel slopes (21.6 versus 55.0 mV dec$^{-1}$) in the deuterium and aqueous media, an inverse KIE is suggested based on $J_H/J_D < 1.0$ (Fig. 4d), indicating that the PCETs are not involved in the RDS. Thus, this result suggests that the catalytic MOR on the in-situ formed NiFe oxyhydroxide is fully exothermic or spontaneous[39].

To probe the precatalytic oxidation process, we studied the H/D isotope effects of NiMn and NiFe-LDH in alkaline $H_2O$ or $D_2O$ without methanol. As shown by their CVs (Supplementary Fig. 19 and Note 9), NiMn-LDH delivers a $Ni^{II}$/$Ni^{III}$ oxidation peak at $^{ox}E_H$/$^{ox}E_D = 1.34/1.33$ $V_{RHE}$ (H and D denote $H_2O$ and $D_2O$, respectively), while NiFe-LDH shows an oxidation peak at $^{ox}E_H$/$^{ox}E_D = 1.44/1.42$ $V_{RHE}$. Based on these experimental results, there are three features to emphasize. First, it is clearly seen that the $Ni^{II}$/$Ni^{III}$ oxidation potential is shifted either cathodically for NiMn-LDH or anodically for NiFe-LDH compared to those of $Ni(OH)_2$ (Supplementary Fig. 20), in line with the literature report[23]. Second, replacing aqueous media by deuterium media leads to easier $Ni^{II}$/$Ni^{III}$ oxidations for both NiMn and NiFe-LDHs. Considering the constant $C_{dl}$ of either LDH, the easier $Ni^{II}$/$Ni^{III}$ oxidation might be attributed to the enhanced adsorption of $OD^-$ with a more polar nature on the $Ni^{II}$ site (Supplementary Fig. 21 and Note 10). Third, it is noted

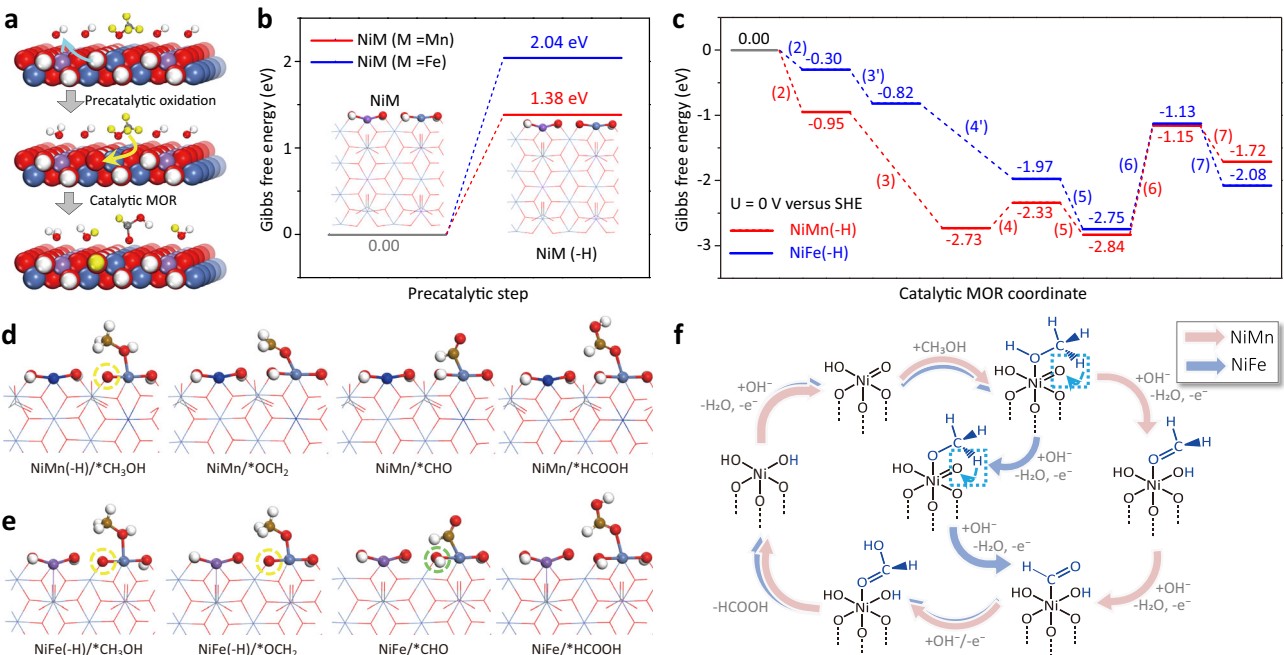

**Fig. 5 | MOR mechanism revealed by experiment and DFT calculation.**
**a** Reaction scheme for the overall MOR on NiM-LDHs (M = Mn, Fe). The blue and yellow arrows indicate the proton-transfers during the precatalytic oxidation process and that associated with the reduction of NiM(-H) to NiM during the catalytic MOR process. The hydrogens of methanol or derived from methanol are highlighted in yellow. Gibbs free energy diagrams for the **b** precatalytic process and

**c** catalytic MOR process. Optimized structures of the intermediates in MOR process on **d** NiMn and **e** NiFe-LDHs, where, Ni: wathet, Fe: purple, Mn: blue, N: gray, O: red, H: white, C: brown. The yellow and green circles indicate the hydrogen-deficient and hydrogen-added oxygens in NiM(-H), respectively. **f** An unconventional bifunctional mechanism proposed for the overall MOR process.

that the $Ni^{II}/Ni^{III}$ oxidation of NiMn-LDH is quasi-irreversible, while that of NiFe-LDH is fully reversible. Based on the above findings, we conclude that the $Mn^{III}$ doping not only facilitates the $Ni^{II}/Ni^{III}$ oxidation but also stabilizes the in-situ formed $Ni^{III}$-OOH, which is beneficial for the following catalytic MOR process. On the other hand, $Fe^{III}$ doping obviously retards the $Ni^{II}/Ni^{III}$ oxidation, which well accounts for why the MOR on the in-situ formed NiFe oxyhydroxide is triggered at a higher onset potential relative to that for NiMn oxyhydroxide.

Based on the experimental results, we propose that the overall MOR of NiM-LDHs (M = Mn, Fe) involves two concomitant processes, including reversible redox transitions of $Ni^{II}$-$(OH)_2$/$Ni^{III}$-OOH and an electrocatalytic MOR (Fig. 5a). DFT computations were then performed to investigate the overall MOR by using two four-layered LDH models of NiM (M = Mn, Fe) (Supplementary Fig. 22a, d)[40-42]. It is noteworthy that the above operando Raman studies have suggested the very limited formation of $Ni^{III}$-OOH for both NiM-LDHs under the MOR conditions. To match the experimental results and also simplify the computational models, we propose a one-electron oxidation (reaction 1)[25] to generate $Ni^{III}$-OOH on the surface of LDH model (Supplementary Fig. 22b, e).

$$NiM + OH^- \rightarrow NiM(-H) + H_2O + e^- \quad (1)$$

The oxidation product (i.e., NiM oxyhydroxide) is denoted as NiM(-H) (M = Mn, Fe), indicating that one hydrogen in NiM-LDH combines with the adsorbed $OH^-$ and leaves as water along with the one-electron oxidation of $Ni^{II}/Ni^{III}$. The calculated Gibbs free energy change ($\Delta G_{OX}$, see Fig. 5b) shows that the precatalytic process (reaction 1) is endothermic for both LDHs, indicating it is thermodynamically unfavorable. Particularly, the $\Delta G_{OX}$ (2.04 eV) of NiFe is much higher than that (1.38 eV) of NiMn. It suggests that a higher oxidation potential is required for NiFe-LDH to achieve $Ni^{III}$-OOH as compared to that of NiMn-LDH, fully consistent with their oxidation behaviors.

To understand the different effects of Mn and Fe dopants on the reaction 1, the projected density of state (PDOS) of Ni-3$d$ electrons was calculated for both LDHs. As shown in Supplementary Fig. 23, the $d$-band center (−2.75 eV) of NiMn is more approaching to the Fermi level as compared to that (−2.98 eV) of NiFe model, indicating the stronger adsorption of $OH^-$ on the surface of NiMn and hence verifying the lower $\Delta G_{OX}$ relative to that of NiFe model.

Afterward, we investigated the reaction pathway of catalytic MOR by using NiMn(-H) as the initial catalyst model. The computations reveal a pathway that begins with the methanol adsorption (reaction 2) on the Ni site of $Ni^{III}$-OOH species and proceeds via a series of dehydrogenations (reactions 3 and 4) to finally form formate (reaction 5):

$$NiM(-H)/* + CH_3OH \rightarrow NiM(-H)/*CH_3OH \quad (2)$$

$$NiM(-H)/*CH_3OH + OH^- \rightarrow NiM/*OCH_2 + H_2O(l) + e^- \quad (3)$$

$$NiM/*OCH_2 + OH^- \rightarrow NiM/*CHO + H_2O(l) + e^- \quad (4)$$

$$NiM/*CHO + OH^- \rightarrow NiM/*HCOOH + e^- \quad (5)$$

where * indicates the active site. It is noteworthy that in reaction 3 the O-H and C-H dehydrogenations synchronously occur along with the one-electron reduction of NiMn(-H) to NiMn (Supplementary Fig. 24). Thus, reaction 3 involves two catalytic sites, which work in a cooperative manner. Particularly, the exposed $Ni^{III}$ provides a site for methanol adsorption, while the nearby electrophilic oxygen acts as a hydrogen acceptor, resulting in a concerted hydrogen-transfer from the adsorbed methanol to NiMn(-H). Thus, these results suggest that the $Ni^{III}$-OOH species provides two active sites for the catalytic MOR on NiMn(-H), which could be defined as the bifunctional mechanism by considering the combined functionalities of active sites.

The corresponding Gibbs energy diagram and the optimized intermediate structures are presented in Fig. 5c, d and Supplementary Fig. 25, respectively. Reactions 2, 3, 5 are exothermic, while reaction 4 is endothermic with $\Delta G = 0.40$ eV. These results are fully consistent with the experimental observations that reveal a spontaneous reduction of $Ni^{III}$-OOH to $Ni^{II}$-$(OH)_2$ under the MOR conditions. Nevertheless, as the reaction 4 provides a thermodynamic barrier, the catalytic MOR process on NiMn(-H) is endothermic or non-spontaneous with the *$OCH_2$ dehydrogenation as the PDS, in line with the H/D KIE study. In addition, we also demonstrate that further electrocatalytic conversion from formate to $CO_2$ via reactions 6 and 7 is thermodynamically unfavorable due to a much higher energy demand ($\Delta G = 1.69$ eV), as compared to that for the formate formation.

$$NiM/*HCOOH + OH^- \rightarrow NiM/*HCOO + H_2O(l) + e^- \quad (6)$$

$$NiM/*HCOO + OH^- \rightarrow NiM/* + CO_2(g) + H_2O(l) + e^- \quad (7)$$

This result is in good agreement with the experimental result, which demonstrates highly selective production of formate instead of $CO_2$. This is very different from that reported for the highly selective methanol-to-$CO_2$ oxidation over NR-$Ni(OH)_2$, where the dehydrogenation of *$CH_2O$ to *CHO is thermodynamically less favored than that of *COOH to *$CO_2$[43].

Alternatively, when using NiFe(-H) as the catalyst model (Supplementary Fig. 22e), the catalytic MOR process was considered with a similar bifunctional mechanism. Nevertheless, if the reaction 3 is applied for NiFe(-H), the following reaction 4 is highly thermodynamically unfavorable with $\Delta G = 1.03$ eV (Supplementary Fig. 26 and Note 11), which is contradict with the KIE study that suggests a fully spontaneous MOR. Alternatively, we propose that, following the O-H dehydrogenation (reaction 3′), the reduction of NiFe(-H) to NiFe occurs along with the double C-H dehydrogenation of methanol (reaction 4′), as listed below.

$$NiM(-H)/*CH_3OH + OH^- \rightarrow NiM(-H)/*OCH_3 + H_2O(l) + e^- \quad (3′)$$

$$NiM(-H)/*OCH_3 + OH^- \rightarrow NiM/*COH + H_2O(l) + e^- \quad (4′)$$

where M represents Fe and the related intermediate structures are provided in Fig. 5e and Supplementary Fig. 27. As a result, the formate formation could be achieved via consecutive exothermic reactions (Fig. 5c). Such a pathway can well account for the fully spontaneous MOR on the NiFe-LDH, as suggested by the H/D KIE study, so it shall have high rationality. Thus, the catalytic MOR on either NiM(-H) can be summarized as the reaction 8.

$$NiM(-H)/* + CH_3OH + 3OH^- \rightarrow NiM/* + HCOOH(l) + 2H_2O(l) + 3e^- \quad (8)$$

To this end, we can verify that the catalytic MOR process is either spontaneous or non-spontaneous depending on the nature of dopant. Nevertheless, considering the high thermodynamic barrier in the precatalytic process, the overall MOR on either LDH is also limited by the $Ni^{II}$-$(OH)_2$/$Ni^{III}$-OOH transition. In another computation, NiM rather than NiM(-H) was used as the initial model for the electrocatalytic MOR, resulting in much higher thermodynamic barriers during the MOR (Supplementary Fig. 28 and Note 12). As it is contradictory with the experimental results, NiM could be ruled out as the origin of catalytic MOR process. In addition, we also conducted computations for OER, which reveal NiFe oxyhydroxide as a better catalyst than NiMn oxyhydroxide (see Supplementary Fig. 29-32 and Note 13). The good

agreement between computations and experiments again justifies the feasibility of our computational method.

Based on the above experimental and computational results, a cyclic pathway is proposed for the overall MOR on the NiM-LDH (Fig. 5f). Briefly, it proceeds via a precatalytic oxidation process to form $Ni^{III}$-OOH species that provides the combined active sites for the MOR. The following process involves a series of methanol dehydrogenations, meanwhile the $Ni^{III}$-OOH is reduced back to the $Ni^{II}$-$(OH)_2$, namely the initial oxidation state in the NiM-LDHs. Along this cyclic pathway, the Ni ions shuttle between $Ni^{II}$ and $Ni^{III}$, while the dopant M (M = Mn, Fe) always remains as $M^{III}$, which, however, exerts influence on not only the $Ni^{II}$-$(OH)_2$/$Ni^{III}$-OOH redox transitions but also the catalytic dehydrogenations of adsorbed methanol, hence determining the catalyst activity. Thus, the data described here constitute the first evidence to support the bifunctional mechanism of MOR, which well accounts for both the reversible redox transitions of $Ni^{II}$-$(OH)_2$/$Ni^{III}$-OOH and the concomitantly occurred MOR that is either spontaneous or non-spontaneous, as observed in the experiments.

## Discussion

In summary, we have reported that the highly selective and continuous production of formate could be achieved at a low cell-potential of 1.33/1.41 V with a current density of 10/100 mA cm$^{-2}$ by using NiMn-LDH instead of NiFe-LDH as the anode catalyst. Through a combined study of operando Raman spectroscopy, H/D KIE survey and DFT computations, we demonstrated a cyclic pathway for the overall MOR, which consists of reversible $Ni^{II}$-$(OH)_2$/$Ni^{III}$-OOH redox transitions and concomitantly occurred MOR. More importantly, we proposed that the key species $Ni^{III}$-OOH provides two combined active sites (i.e., $Ni^{III}$ and nearby electrophilic oxygen) to promote the MOR, which defined as the bifunctional mechanism can well account for both the highly selective formation of formate and the transient presence of $Ni^{III}$-OOH during the MOR. Moreover, the different catalytic performance of NiMn and NiFe-LDHs could be attributed to their different $Ni^{II}$-$(OH)_2$/$Ni^{III}$-OOH oxidation behaviors, which trigger either non-spontaneous or spontaneous MOR process. Thus, our work presents a clear understanding on the MOR mechanism of nickel-based hydroxides, which could be expanded to encompass the electro-oxidations of various primary and secondary alcohols that have at least one hydrogen atom on the carbon attached to the hydroxyl group. The survey on the bifunctional mechanism of MOR provides a new principle for catalyst design in the field of electro-oxidations of alcohols.

## Methods

### Materials
Details for materials can be found in Supplementary Note 14.

### Catalyst synthesis
The preparation of NiMn-LDH on Ni foam (NiMn-LDH/NF) was used by a modified hydrothermal method. First, $Ni(NO_3)_2 \cdot 6H_2O$ (0.24 mmol), $KMnO_4$ (0.06 mmol) and $CO(NH_2)_2$ (2.00 mmol) were added to deionized water (15 mL) and stirred for 20 min. A piece of NF (2 × 3 cm$^2$) was ultrasonically cleaned with diluted HCl solution (2 M), ethanol and deionized water for 15 min respectively, and then dried for use. Then, the reaction mixture was transferred to a Teflon-lined autoclave (35 mL) with a piece of cleaned NF and treated at 120 °C for 12 h. After cooling to room temperature, the NiMn-LDH/NF was rinsed with deionized water, ethanol each for 3 times, and dried at 60 °C overnight. Alternatively, NiFe-LDH/NF was synthesized under the same conditions by using $Ni(NO_3)_2 \cdot 6H_2O$ (0.24 mmol) and $Fe(NO_3)_3 \cdot 9H_2O$ (0.06 mmol) as the precursors. To synthesize $Ni(OH)_2$/NF, only $Ni(NO_3)_2 \cdot 6H_2O$ (0.30 mmol) was used as the precursor. To optimize the M content (M = Mn, Fe) in NiM-LDH, the molar ratio of $Ni^{2+}$ and $M^{x+}$ precursors was varied from 0.95:0.05 to 0.90:0.10, 0.85:0.15, 0.80:0.20, 0.75:0.25, meanwhile the total molar amount of precursors

were kept constant. The synthesized products could be denoted as Ni$_{1-y}$M$_y$-LDH, where $y$ represents the relative molar content of M$^{x+}$ precursor. The optimized precursors ratio could be determined by checking their MOR activities (Supplementary Fig. 33 and Note 15).

## Characterization

Scanning electron microscopy (SEM), transmission electron microscopy (TEM) images and high-resolution TEM (HR-TEM) images along with element analysis mapping were recorded using Hitachi S-8010 and Titan Themis Cubed G2300 instruments, respectively. The catalysts grown on NF are used as the SEM samples. To prepare the TEM sample, the catalyst was removed from NF by sonication in ethanol solution for 1 h and a drop of supernatant was deposited on a duplex Cu mesh. X-ray diffraction (XRD) patterns were measured using a Bruker D8 Advance instrument with a Cu Kα radiation source (λ = 1.54178 Å). The XRD samples were synthesized without the addition of NF in the reactor. X-ray photoelectron spectroscopy (XPS) measurements were conducted with a Thermo Fisher Escalab 250Xi instrument. The catalysts grown on NF (see Supplementary Fig. 1) are used as the XPS samples. Inductively coupled plasma (ICP) was conducted on an OPTIMA 8000 instrument. The sample for ICP measurement was prepared by dissolving the catalyst in concentrated hydrochloric acid. The formate concentration in electrolyte was detected by a Thermo Scientic Dionex ion chromatography (IC) system. $^1$H and $^{13}$C nuclear magnetic resonance (NMR) spectra were recorded with a BRUKER AVANCEIII HD 400 MHz NMR instrument by using D$_2$O as solvent and methanol (CH$_3$OH) as the internal reference.

## Electrochemical measurements

Details for electrochemical measurements and calculations of Faradaic efficiency can be found in Supplementary Fig. 34, 35 and Note 16, 17.

## Operando Raman measurements

The Operando Raman measurements were implemented in a confocal Raman microscope (HR Evolution, Horiba Jobin Yvon) with a 50x objective and the excitation source of laser wavelength of 633 nm. Charge coupled device (CCD) detector was working at −60 °C to collect the scattered light from sample surface. Spectral shifts were calibrated based on the value of 520.7 cm$^{-1}$ of a silicon wafer. Operando Raman spectra were acquired under controlled potentials using a tailor-made Teflon cell connected to a CHI 760E electrochemical workstation. The as-prepared catalysts loaded on NF, graphite rod and Ag/AgCl (Saturated KCl) were served as working, counter and reference electrode, respectively. The electrochemical-Raman measurements were executed by chronopotentiometry.

## Computational methods

Details for computational methods can be found in Supplementary Note 18.

# Data availability

The experimental data generated in this study are provided in the Supplementary Information/Source Data file. Additional data are available from the corresponding author upon reasonable request. Source data are provided with this paper.

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

## Acknowledgements

This work was supported in part by the National Key Research and Development Program of China (2020YFB1506401, L.F.), the National Natural Science Foundation of China (Grant no. 52172050, L.F. and 22171068, P.J.), Natural Science Research Project of Jiangsu Higher Education Institutions (21KJA480002, L.F.), six talent peaks project in Jiangsu province (XCL-078, L.F.) and Suzhou Key Laboratory for Advanced Carbon Materials and Wearable Energy Technologies. We also thank for the support from Soochow Municipal laboratory for low carbon technologies and industries.

## Author contributions

B.T.Z. and B.D. performed the main experiments. F.W., Q.F.Y., and Y.P.H. are responsible for part of experiments and analysis. B.T.Z., C.J.Z., and P.J. are responsible for DFT calculations. B.T.Z., P.J., and L.F. designed the project, analyzed the results and wrote the manuscript.

## Competing interests

The authors declare no competing interests.
