## [Peer review file · Nature Communications]

REVIEWER COMMENTS

Reviewer #1 (Remarks to the Author):

The methanol oxidation reaction at nickel oxyhydroxide based electrodes in base has been studied for many years. The present paper using NiMn and NiFe LDH materials adds significant value to this literature and will be read with considerable interest by other workers in this field. The methodology adopted by the authors is sound. The methodology section contains good detail. The paper is well written and the quality of the data is excellent. I particularly like the kinetic and detailed mechanistic analysis presented. The results obtained with respect to MOR electrocatalysis are very promising. The nice use of in situ Raman are noteworthy. The blend of computational analysis and clear experimental results is very nice. Fig.5 is very helpful. For these many reasons I recommend publication for this important paper.

I have one small point. The concept of threshold potential is purely empirical and has no real physical basis. When comparing catalytic activity it is better to stick to quoting the overpotential at a fixed current density in the Tafel region.

Reviewer #2 (Remarks to the Author):

This work reports a comprehensive study of the methanol oxidation reaction (MOR) with Ni-based hydroxides as electrocatalyst materials, namely NiMn and NiFe hydroxides. In general terms, this is a relevant field in electrocatalysis as alcohol oxidation can be extremely useful in both energy conversion technologies and for sustainable chemical synthesis. Overall, I think this is an excellent work that provides fresh mechanistic understanding of the electrocatalytic MOR using earth-abundant metal hydroxides. The authors proposed a well-planned set of experiments, supported by computational modelling, to elucidate a mechanism that has been in discussion for the past few decades. Data provided by operando Raman and the studies with deuterated chemicals (Raman, and kinetic isotope effect) are key to reach this new understanding and provide a full picture of MOR reactivity.

Not only these results are interesting from a fundamental perspective for mechanistic determination, but the long-term electrolysis (20h) at 100 mA cm^{-2} , and the high selectivity to formate demonstrates the excellent practical performance of the NiMn electrocatalyst for the MOR. Indeed, an important challenge using noble metal-free electrocatalysts for alcohol oxidation is the need to apply potentials close to the oxygen evolution reaction (OER), which impedes to increase the current densities without decreasing the faradaic efficiency towards the reaction of interest. The large difference in activity between the OER and the MOR for the NiMn electrocatalyst enables this possibility as it is demonstrated in this paper.

In short, I have really enjoyed reading this paper and I am enthusiastic for its publication. This work has the potential to make an impact in this particular field of electrocatalysis, as I envision that mechanistic considerations in this work might also be applicable to other alcohol oxidation reactions involving Ni-based hydroxides. While I recommend the publication in Nature Communications, I think there are a few aspects that the authors could improve or clarify before the paper is published:

- My major concern is about the strength of the data from an analytical and statistical point of view as it looks like every figure and calculation only shows data from one single experiment without any replicate. I understand repeating every single experiment in the paper is unfeasible, but at least showing the LSV response obtained for the MOR with several independent electrodes might indicate how reproducible is the preparation of these materials and the electrochemical performance.
- There are a few mentions of the bifunctional mechanism in the Introduction. It might be a good idea to briefly describe what a bifunctional mechanism is in the introduction, particularly for the broad readership of the journal, which are not expected to be experts in this topic.
- This is the first time I have come across the term nucleophile oxidation reaction (NOR) for this type of electrocatalytic reactions. I do not dislike the term, but I guess this is a recent addition to the nomenclature in electrocatalysis as nucleophile is more widely used term in organic chemistry. Could authors describe what they refer to by this expression when it is mentioned in the introduction?
- Authors describe in the Introduction a previously proposed mechanism by generation of O* or OH* species. It might be a good idea to include in the discussion how the newly proposed mechanism differs from that one to provide more context about the differences with previously proposed mechanisms for this reaction.
- Authors could also further define the spontaneous vs non-spontaneous methanol activation idea, as it is not very clear what they mean by activation in this context. Could they refer to the initial adsorption of the reactant on the catalyst as the first step for the whole process?
- It is mentioned that the equilibrium potential (I guess standard potential would be a better term?) of the methanol oxidation is 1.0 V vs RHE (Line 48). Could authors check and confirm this value? I doubt very much this is the correct value as oxidation of methanol is possible at a lower potential using noble-metal electrocatalysts. The MOR to CO₂ is, for instance, ca. +0.02 V vs SHE.

- “Before the electrochemical tests, the cyclic voltammetry (CV) was tested until a constant curve was achieved”. Could you provide information about the specific parameters for this experiment (potential range, scan rate, etc.)?
- Could you clarify if the iR compensation was carried out manually after the experiment or directly by the potentiostat?
- Just a note of advice: the Ag/AgCl reference electrode is not the best electrode for experiments in alkaline media, and a Hg/HgO electrode would be more appropriate. I do not think the results would be much different, but this fact could be relevant for other referees that you encounter in the future. The choice of a PTFE cell instead of glass shows that you have really thought about the effect of alkaline media in other experimental aspects.
- Line 32: I guess authors mean “high-energy density” instead of “high-density energy”.
- Line 255 in SI: small typo – reversible instead of “riverside”.
- Was any pretreatment carried out to the Nafion membrane when used in the two-compartment cell such as wetting in a specific solution? This information is missing.
- A specific capacitance of $60 \mu\text{F cm}^{-2}$ was used to calculate the ECSA for the different catalysts. Have you considered that this value might change depending on the composition of the electrocatalyst?
- Line 267 in SI: It seems like the FE for formate was calculated using the equation described there, which only considers the electrochemical charge. However, it is mentioned in the manuscript that the FE is calculated after quantifying the amount of formate produced, which is the correct way to consider possible side reactions. Could authors update this part of the SI to include the full method used for FE calculations?
- Could authors provide further information about how the samples for the different characterisation experiments were prepared? For instance, there is not information about the sample preparation for TEM, XRD or XPS. There is not contribution from the Ni foam in these analysis, so I assume the catalyst layer is removed from the surface beforehand.

- Could authors tentatively assign the different peaks of the XRD pattern to the specific crystalline phases?
- Line 116: “The smaller Tafel slope indicates faster MOR kinetics on NiMn than on NiFe-LDH”. This is true if they follow the same mechanism and share similar rate-determining steps. However, I wonder if the different MOR mechanisms proposed for these catalysts might affect the Tafel slope obtained experimentally.
- Figure 2C, ECSA: It is not clear in the first instance the presence of two y-axes. I suggest to modify this figure in order to better differentiate the different data.
- There is a mention to EWS abbreviation in Figure S9, but it is not defined anywhere. Is this for electrochemical water splitting? If so, this has been defined as overall water splitting (OWS) in the manuscript.
- Could authors include a few brief sentences about the potential of this work to understand further alcohol oxidation reactions with this type of electrocatalysts? How they envision these results are relevant to other reactions beyond methanol?
- I am not sure “deuterio” is the correct term. I would stick the use to “deuterium”, but I could be mistaken.
- Line 191: “Methanol-induced reduction of Ni^{III}-OOH was suppressed due to the more robust O-D bond”. Could authors further discuss this idea? Do they mean the OD bond in methanol or in the oxyhydroxide? I would also not say “suppressed” as the Raman bands do not seem as sharp and intense as in the OER case, so there might be still some reaction going as the KIE experiments also indicate.
- Do the experiments with deuterium are carried out after some equilibration time or the exchange between H and D in NiOH₂ is instantaneous after immersing into the solution?
- How was the preparation of the D₂O? Having well-purified D₂O seems to be essential for good electrocatalytic studies such as reported in <https://doi.org/10.1038/s41557-022-01084-y>

- Figure S18 for NiMn in D₂O: the Ni processes show two distinctive peaks, with the second one at about 1.45 V. What is the origin of this process? It looks like it does not appear in the same experiment carried out in H₂O.
- Would the authors be able to discuss how the use of D₂O could vary the electrical double layer or the potential of zero charge? Could these phenomena be partially responsible for the differences observed for the Ni(II)/Ni(III) processes between aqueous and deuterium media?
- Line 272: “FeIII doping suppresses the Ni(II)/Ni(III) oxidation”. I do not think suppresses is the right word as the process is still happening but at a higher potential.
- Were the LSV curves recorded in deuterium media also compensated by iR drop? Does the use of D₂O changes the uncompensated resistance?

Reviewer #3 (Remarks to the Author):

The authors explored a bifunctional mechanism of methanol-to-formate electrooxidation on NiFe/NiMn-based hydroxides. The results reveal that Mn-doped NiOOH NiMn-LDH has a better catalytic activity than the NiFe-LDH based anodic catalyst. However, I do not recommend this paper can be published in Nature Communications since no insights into NiOOH-based catalytic mechanism. Specific issues to be addressed are:

No new insights can be found in this work. Mn and Fe-embedded NiOOH-based catalysts have been well reported. Also, why NiMn-LDH has a better catalytic performance than NiFe-LDH? The insight into the catalytic mechanism is essential for the design of the catalyst.

The concentration of Fe or Mn significantly affects the overpotential and reaction kinetics?

Since the active site is Ni, how embedded Mn or Fe contributes to the electrochemical reaction differently?

The Gibbs free energy should be converged to 4.92 eV (4x1.23) shown in Figures S26 and S27? This needs to be clarified.

REVIEWER COMMENTS

Reviewer #1 (Remarks to the Author):

The methanol oxidation reaction at nickel oxyhydroxide based electrodes in base has been studied for many years. The present paper using NiMn and NiFe LDH materials adds significant value to this literature and will be read with considerable interest by other workers in this field. The methodology adopted by the authors is sound. The methodology section contains good detail. The paper is well written and the quality of the data is excellent. I particularly like the kinetic and detailed mechanistic analysis presented. The results obtained with respect to MOR electrocatalysis are very promising. The nice use of in situ Raman are noteworthy. The blend of computational analysis and clear experimental results is very nice. Fig.5 is very helpful. For these many reasons I recommend publication for this important paper.

I have one small point. The concept of threshold potential is purely empirical and has no real physical basis. When comparing catalytic activity it is better to stick to quoting the overpotential at a fixed current density in the Tafel region.

Response: Thanks a lot for this professional comment. According to the comment 6 by reviewer 2#, we have corrected the standard potential of MOR (methanol to formate oxidation reaction) to 0.103 V (vs SHE). To response this comment, we calculated the onset overpotentials for the MOR on NiMn and NiFe-LDHs as follows.

“NiMn-LDH exhibits a remarkable lower MOR onset potential (at 2 mA cm⁻²) of 1.30 V_{RHE} (or a smaller onset overpotential of 1.20 V) as compared to that (1.37 V_{RHE} or 1.27 V) of NiFe-LDH, indicating that NiMn-LDH can more efficiently boost MOR under the applied conditions (Fig. 2a, Supplementary Fig. 6 and note 3).”

Reviewer #2 (Remarks to the Author):

This work reports a comprehensive study of the methanol oxidation reaction (MOR) with Ni-based hydroxides as electrocatalyst materials, namely NiMn and NiFe hydroxides. In general terms, this is a relevant field in electrocatalysis as alcohol oxidation can be extremely useful in both energy conversion technologies and for sustainable chemical synthesis. Overall, I think this is an excellent work that provides fresh mechanistic understanding of the electrocatalytic MOR using earth-abundant metal hydroxides. The authors proposed a well-planned set of experiments, supported by computational modelling, to elucidate a mechanism that has been in discussion for the past few decades. Data provided by operando Raman and the studies with deuterated chemicals (Raman, and kinetic isotope effect) are key to reach this new understanding and provide a full picture of MOR reactivity.

Not only these results are interesting from a fundamental perspective for mechanistic determination, but the long-term electrolysis (20h) at 100 mA cm⁻², and the high selectivity to formate demonstrates the excellent practical performance of the NiMn electrocatalyst for the MOR. Indeed, an important challenge using noble metal-free electrocatalysts for alcohol oxidation is the need to apply potentials close

to the oxygen evolution reaction (OER), which impedes to increase the current densities without decreasing the faradaic efficiency towards the reaction of interest. The large difference in activity between the OER and the MOR for the NiMn electrocatalyst enables this possibility as it is demonstrated in this paper.

In short, I have really enjoyed reading this paper and I am enthusiastic for its publication. This work have the potential to make an impact in this particular field of electrocatalysis, as I envision that mechanistic considerations in this work might also be applicable to other alcohol oxidation reactions involving Ni-based hydroxides. While I recommend the publication in Nature Communications, I think there are a few aspects that the authors could improve or clarify before the paper is published.

Comment 1: My major concern is about the strength of the data from an analytical and statistical point of view as it looks like every figure and calculation only shows data from one single experiment without any replicate. I understand repeating every single experiment in the paper is unfeasible, but at least showing the LSV response obtained for the MOR with several independent electrodes might indicate how reproducible is the preparation of these materials and the electrochemical performance.

Response 1: Thanks a lot for this useful comment. To response this comment, we have repeated the LSV and C_{dl} measurements by using three independent electrodes per each catalyst. The LSV curves of NiMn and NiFe-LDH modified electrodes towards MOR are provided in **Supplementary Fig. 6a**. It is clearly seen that the LSV curves for each catalyst are almost overlapped with each other, indicating that both the catalyst preparations and electrochemical measurements are reproducible. We also repeated the CV measurements to estimate the C_{dl} values of these individual electrodes. As shown in **Supplementary Fig. 7**, the C_{dl} values calculated for three individual electrodes per each catalyst are very close to each other, indicating again that both the catalyst preparations and electrochemical performance are reproducible.

Supplementary Fig. 6. (a) LSV curves (with iR compensation) of NiMn and NiFe-LDH (including three independent electrodes per each catalyst) measured with

scan rate of 5 mV s^{-1} under the MOR conditions (in 1 M KOH with 3 M methanol).

Supplementary note 3

To explore the reproducibility of the catalyst performance, we have repeated the LSV measurements under the MOR conditions by using three independent electrodes per each catalyst. As shown in **Supplementary Fig. 6a**, the LSV curves for each catalyst are almost overlapped with each other, indicating that both the catalyst preparations and electrochemical measurements are reproducible.

Supplementary Fig. 7. Calculation of C_{dl} : CV curves of (a-c) NiMn-LDH and (d-f) NiFe-LDH (totally three independent electrodes per each catalyst) measured in the non-Faradaic range under the same conditions. Plots of current density and scan rate (ν) of (g) NiMn-LDH and (h) NiFe-LDH. Average values of C_{dl} could be determined as 3.23 mF (NiMn-LDH) and 3.37 mF (NiFe-LDH) for the electrode with an area of 1 cm^2 .

Supplementary note 4

To explore the reproducibility of the C_{dl} , we have repeated the CV measurements under the same conditions by using three independent electrodes per each catalyst. As shown in **Supplementary Fig. 7g,h**, the plots of current density and scan rate of each catalyst are very close to each other, indicating that the C_{dl} measurements are reproducible. For each catalyst, an averaged value was adopted to determine the C_{dl} .

Comment 2: There are a few mentions of the bifunctional mechanism in the Introduction. It might be a good idea to briefly describe what a bifunctional mechanism is in the introduction, particularly for the broad readership of the journal, which are not expected to be experts in this topic.

Response 2: Thanks a lot for this valuable comment. To response this comment, we have defined the concept of bifunctional mechanism at the end of introduction section, which is based on the literature definition for the bifunctional mechanism of OER (*Angew. Chem. Int. Ed.* **2021**, 60, 3095–3103).

“Furthermore, based on the experimental results, density functional theory (DFT) computations suggest a cyclic pathway consisting of reversible $\text{Ni}^{\text{II}}/\text{Ni}^{\text{III}}$ redox transitions and a concomitant MOR. More importantly, it is verified that the electrocatalytic MOR involves two active sites including Ni^{III} and nearby electrophilic oxygen species of $\text{Ni}^{\text{III}}\text{-OOH}$, which work in a cooperative manner to promote the MOR. Such a mechanism is different from that based on a single active site and hence denoted as the bifunctional mechanism by considering the combined functionalities of both active sites²⁶”

Comment 3: This is the first time I have come across the term nucleophile oxidation

reaction (NOR) for this type of electrocatalytic reactions. I do not dislike the term, but I guess this is a recent addition to the nomenclature in electrocatalysis as nucleophile is more widely used term in organic chemistry. Could authors describe what they refer to by this expression when it is mentioned in the introduction?

Response 3: Thanks a lot for this valuable comment. To response this comment, we have modified the introduction, in which the concept of nucleophile in electro-catalysis is provided as follows, based on the literature definition.

“Nickel-based anode catalysts have been widely used for OER and various biomass electro-oxidations. As many of biomass (i.e., methanol, ethanol, furfural, and furfurylamine) have a nucleophilic group (i.e., hydroxyl, aldehyde, and amino groups), their oxidations are defined as nucleophile oxidation reaction (NOR)¹⁶⁻²⁰.”

Comment 4: Authors describe in the Introduction a previously proposed mechanism by generation of O* or OH* species. It might be a good idea to include in the discussion how the newly proposed mechanism differs from that one to provide more context about the differences with previously proposed mechanisms for this reaction.

Response 4: Thanks a lot for this valuable comment. To response this comment, we have modified the introduction section and provided more clear introduction for previously proposed EOR and MOR mechanisms on Ni(OH)₂ (*Chem* **2020**, *6*, 2974-2993), NiFeO_x (*Adv. Energy Mater.* **2021**, *11*, 2101858) and Ni(OH)₂ (*Appl. Catal., B* **2021**, *281*, 119510), respectively. All these catalysts are Ni-based hydroxide or oxide, which make them comparable to our catalysts including NiMn and NiFe-LDH. The literatures reported mechanisms are described as follows:

*“To more efficiently produce value-added products from biomass, many efforts have been devoted to understanding the NOR process²¹⁻²⁵. In a state-of-the-art work, Wang et al. focused on the change of Ni(OH)₂ catalyst under the EOR conditions and proposed a one-electron reaction consisting of an endothermic oxidation of Ni^{II}-(OH)₂ to Ni^{III}-OOH and an exothermic or “spontaneous” reduction of Ni^{III}-OOH to Ni^{II}-(OH)₂ along with the dehydrogenation of ethanol²³. Subsequently, Shalom et al. surveyed on the electrocatalytic process of MOR and revealed that the overall rate of MOR is bottlenecked by the C-H bond oxidation of methanol, rather than the formation of Ni^{III}-OOH on the surface of NiFeO_x²⁴. Another group also afforded similar results based on an uphill energy barrier upon the conversion from *OCH₂ to *OCH during the MOR of Ni(OH)₂ with Ni^{III}-OOH species²⁵. Thus, the in-situ formed Ni^{III}-OOH was generally believed as the origin of MOR or EOR. However, experimental identification was blocked due to the transient stability of Ni^{III}-OOH especially under optimal MOR or EOR conditions (i.e., at a potential of < 1.5 V_{RHE} with a high alcohol concentration). To date, how this transient species promotes the alcohol oxidation remains ambiguous. The most typical mechanism proposed the electrophilic oxygen of Ni^{III}-OOH as the single active site towards MOR²⁵, which may well account for the selective formation of formate but not the rapidly diminishing of Ni^{III}-OOH during the MOR. To further address both phenomena, an alternative*

catalytic mechanism is required.”

At the end of introduction section, we highlight the features of our proposed bifunctional mechanism, which involves **two combined active sites** rather than the **single active site** proposed by the literature. Moreover, as the bifunctional mechanism can well account for not only the selective formation of formate but also the transient presence of Ni^{III}-OOH, as observed in experiments, it shall be more rational than the previous proposed mechanism involving only a single active site. The modified discussion is as follows:

“Furthermore, based on the experimental results, density functional theory (DFT) computations suggest a cyclic pathway consisting of reversible Ni^{II}/Ni^{III} redox transitions and a concomitant MOR. More importantly, it is verified that the electrocatalytic MOR involves two active sites including Ni^{III} and nearby electrophilic oxygen species of Ni^{III}-OOH, which work in a cooperative manner to promote the MOR. Such a mechanism is different from that based on a single active site and hence denoted as the bifunctional mechanism by considering the combined functionalities of both active sites²⁶”

One more point is that the earlier literature works (*Chem* **2020**, *6*, 2974-2993, and *Adv. Energy Mater.* **2021**, *11*, 2101858) proposed EOR (ethanol oxidation reaction) and MOR mechanism based on experimental results only, which makes the determination of active site very difficult. In our work, we combined experiments and computations together, through which not only the active sites could be clarified but also the rationality of the results could be mutually verified.

Another note is that mechanism of the MOR on Ni/NiB_x heterostructure (*Nat. Commun.* **2022**, *13*, 4602) is no longer compared with our work in the introduction section, as the catalyst with a Ni/NiB_x heterostructure is very different from our catalysts based on nickel-based hydroxides. Although there is the formation of Ni-OOH on the Ni surface of Ni/NiB_x, the metallic Ni is also proposed as the active site. It is very different from the MOR or EOR on the nickel based hydroxides or oxides, which have no metallic nickel species.

Comment 5: Authors could also further define the spontaneous vs non-spontaneous methanol activation idea, as it is not very clear what they mean by activation in this context. Could they refer to the initial adsorption of the reactant on the catalyst as the first step for the whole process?

Response 5: Thanks a lot for this valuable comment. Actually, the terms of “spontaneous and non-spontaneous” refer to the whole process of MOR, which is either exothermic or endothermic. To clarify this, we have defined the term “spontaneous” as “exothermic” and “non-spontaneous” as endothermic in the main text. In addition, to avoid the confusing, we did not use the phrase “methanol activation” in the revised version, we used the phrase “spontaneous or non-spontaneous MOR”, in which the term MOR has been defined as the whole

process of methanol-to-formate electro-oxidation reaction in the introduction.

For example:

“Thus, this result suggests that the catalytic MOR on the in-situ formed NiFe oxyhydroxide is fully exothermic or “spontaneous”³⁹.”

Comment 6: It is mentioned that the equilibrium potential (I guess standard potential would be a better term?) of the methanol oxidation is 1.0 V vs RHE (Line 48). Could authors check and confirm this value? I doubt very much this is the correct value as oxidation of methanol is possible at a lower potential using noble-metal electrocatalysts. The MOR to CO₂ is, for instance, ca. +0.02 V vs SHE.

Response 6: Thanks a lot for this very professional comment. To clarify the equilibrium potential and standard potential, we have checked the book, "Electrochemical Dictionary" (edited by Allen J. Bard, György Inzelt, Fritz Scholz; 2008 Springer-Verlag Berlin Heidelberg). The standard potential (E^\ominus) could be calculated by using a thermodynamic equation of $E^\ominus = -\Delta G^\ominus/nF$, where n is the number of transferred electrons, $-\Delta G^\ominus$ is the standard Gibbs energy, and F is the Faraday constant. As the MOR involves 4 electrons and a ΔG^\ominus of 40.1 kJ mol⁻¹ (*Angew. Chem. Int. Ed.* **2021**, *60*, 3148-3155), the E^\ominus could be calculated as 0.103 V (vs SHE). If using RHE as the reference electrode, the E^\ominus could be 0.103 V (vs RHE) regardless of the pH of electrolyte.

So, we correct the term “equilibrium potential” to “standard potential” in the introduction section as follows.

“Nevertheless, the OWS normally requires a cell-potential higher than 1.6 V due to the sluggish oxygen evolution reaction (OER) with a high standard potential of 1.23 V_{SHE} at the anodic side^{4,5}”

“Alternatively, methanol electro-oxidation reaction (MOR) to formate has been of great interest very recently due to the mild synthesis condition and low standard potential of 0.103 V_{SHE}^{5,10,11}”

Comment 7: “Before the electrochemical tests, the cyclic voltammetry (CV) was tested until a constant curve was achieved”. Could you provide information about the specific parameters for this experiment (potential range, scan rate, etc.)?

Response 7: Thanks a lot for this professional comment. To response this comment, we have added more details regarding the conditions of CV activation in the revised SI.

Supplementary note 16

Electrochemical measurements: *“The CV activations for all catalysts were*

conducted at a scan rate of 50 mV s^{-1} in the potential range of 1.02-1.62 V_{RHE} in the corresponding electrolyte.”

Comment 8: Could you clarify if the iR compensation was carried out manually after the experiment or directly by the potentiostat?

Response 8: Thanks a lot for this professional comment. In fact, the iR compensation was carried out manually after the LSV measurements. We have provided corresponding details in the revised SI as follows. It is also noted that the LSV curve of NiMn-LDH without iR compensation have been provided in **Supplementary Fig. 6b** for comparison.

Supplementary note 16

Electrochemical measurements: "In addition, for all LSV curves, 90% iR drop compensation was applied manually after the experiment, where R was notarized by fitting electrochemical impedance spectroscopy (EIS) data."

Supplementary Fig. 6. (b) Comparison between the LSV curves of NiMn-LDH with and without iR compensation, where $R = 1.65 \text{ Ohm}$.

Comment 9: Just a note of advice: the Ag/AgCl reference electrode is not the best electrode for experiments in alkaline media, and a Hg/HgO electrode would be more appropriate. I do not think the results would be much different, but this fact could be relevant for other referees that you encounter in the future. The choice of a PTFE cell instead of glass shows that you have really thought about the effect of alkaline media in other experimental aspects.

Response 9: Thanks a lot for this professional comment. In fact, we used the Ag/AgCl electrode with a **double salt bridge** (see photo below) for all electrochemical tests. As the double salt bridge may prevent the damage of alkaline electrolyte to the Ag/AgCl electrode, the Ag/AgCl electrode with a **double salt bridge** could be used in alkaline media. Alternatively, the Hg/HgO electrode is more suitable for alkaline media. To compare the results with different reference electrodes, we performed the LSV

measurements again by using the Hg/HgO as reference electrode. The LSV curves obtained with Ag/AgCl and Hg/HgO reference electrodes are provided in the **Figure for response 9**. It is clearly seen that these LSV curves are almost the same, regardless of the reference electrodes. In the future works, we will use Hg/HgO reference electrode for the electrochemical tests in alkaline media.

Photo showing the Ag/AgCl reference electrode with a **double salt bridge** used for our electrochemical measurements.

Figure mentioned in Response 9. LSV curves (without iR compensation) of NiMn and NiFe-LDH measured under the MOR conditions using Ag/AgCl with a double salt bridge or Hg/HgO as the reference electrode.

Comment 10: Line 32: I guess authors mean “high-energy density” instead of “high-density energy”.

Response 10: Thanks a lot for pointing out this error. To response this comment, we have corrected this phrase in the revised manuscript.

"Hydrogen (H₂) as a carbon-free and high-energy density fuel has been considered as one of the most important alternatives to conventional fossil fuels¹."

Comment 11: Line 255 in SI: small typo – reversible instead of “riverside”.

Response 11: Thanks a lot for pointing out this error. To response this comment, we have corrected this word in the revised SI.

Comment 12: Was any pretreatment carried out to the Nafion membrane when used in the two-compartment cell such as wetting in a specific solution? This information is missing.

Response 12: Thanks a lot for this professional comment. In fact, the Nafion membrane has been pretreated before use. We provide details for the pretreatment in the SI.

Supplementary note 16

“To perform the HER/MOR electrolysis, a two-electrode system was set up by using a single-chamber electrolyzer or a double-chamber electrolyzer (H-cell) with a Nafion 117 membrane as the separator. Before setting up, the Nafion membrane was first treated in 5 wt% hydrogen peroxide at 80 °C for 1h, then soaked in deionized water for 30 min; and then boiled in 5 wt% dilute sulfuric acid at 80 °C for 1 h; and finally soaked in deionized water for 30 min.”

Comment 13: A specific capacitance of 60 $\mu\text{F cm}^{-2}$ was used to calculate the ECSA for the different catalysts. Have you considered that this value might change depending on the composition of the electrocatalyst?

Response 13: Thanks a lot for this professional comment. It is true that the specific capacitance (C_s) might change depending on the composition of the electrocatalyst. To response this comment, we adopt the method reported in the literature (*Appl. Catal. B: Environ.* **2020**, 265, 118543), C_s value is determined from independent experiment utilizing known-area electrode according to the following equation:

$$C_s = \frac{\int_{V_1}^{V_2} I dV}{2v \Delta V}$$

Where $\int_{V_1}^{V_2} I dV$ is the area of CVs, v is scan rate (V s^{-1}), and ΔV is potential window. Thus, C_s is the slope of scan rate and $\frac{\int_{V_1}^{V_2} I dV}{2\Delta V}$ plot.

We, therefore, performed the calculations for NiMn and NiFe-LDH, respectively, based on which average C_s value of 2.65 mF cm^{-2} (for NiMn-LDH) or 2.89 mF cm^{-2} (for NiFe-LDH) was obtained by using three independent electrodes for each catalyst. The calculation details and results are provided in **Supplementary Fig. 8**. Accordingly, the ECSA was calculated to be 1.22 cm^2 for NiMn-LDH and 1.17 cm^2 for NiFe-LDH using the equation of $\text{ECSA} = C_{dl}/C_s$. The details for C_s calculations are

provided in **Supplementary note 16**.

“As the C_s may change depending on the composition of the electrocatalyst, it is determined from independent experiment utilizing known-area electrode according to the following equation¹²:

$$C_s = \frac{\int_{V_1}^{V_2} I dV}{2v \Delta V}$$

Where $\int_{V_1}^{V_2} I dV$ is the area of CVs, v is scan rate ($V s^{-1}$), and ΔV is potential window.

Thus, C_s is the slope of scan rate and $\frac{\int_{V_1}^{V_2} I dV}{2\Delta V}$ plot.”

Supplementary Fig. 8. Calculation of C_s : Plots of scan rate (v) and $\frac{\int_{V_1}^{V_2} I dV}{2\Delta V}$ for (a) NiMn-LDH and (b) NiFe-LDH (totally three independent electrodes per each catalyst). Average values of C_s are determined as 2.65 mF cm⁻² (NiMn-LDH) and 2.89 mF cm⁻² (NiFe-LDH). As a result, $ECSA$ could be determined as 1.22 cm² (NiMn-LDH) and 1.17 cm² (NiFe-LDH) based on the equation of $ECSA = C_{dl}/C_s$.

Comment 14: Line 267 in SI: It seems like the FE for formate was calculated using the equation described there, which only considers the electrochemical charge. However, it is mentioned in the manuscript that the FE is calculated after quantifying the amount of formate produced, which is the correct way to consider possible side reactions. Could authors update this part of the SI to include the full method used for FE calculations?

Response 14: Thanks a lot for this useful comment. Yes, we did calculate the FE for formate after quantifying the amount of formate produced by IC. And we have updated the full method used for FE calculations in **Supplementary note 17** in the revised SI:

“**Calculation method for FE:** The generated formate at the anode was detected by

ion chromatography (IC). The Faraday efficiency (FE) of formate can be calculated using the following equation:

$$FE(\text{formate}) = \frac{n \times 4 \times N_A \times e}{Q} \times 100\%$$

where n is the mol of generated formate; 4 is the number of transferred electrons; N_A is Avogadro constant ($6.02 \times 10^{23} \text{ mol}^{-1}$); e is elementary charge ($1.60 \times 10^{-19} \text{ C}$); Q is the passed charge (C)."

Comment 15: Could authors provide further information about how the samples for the different characterization experiments were prepared? For instance, there is not information about the sample preparation for TEM, XRD or XPS. There is not contribution from the Ni foam in these analysis, so I assume the catalyst layer is removed from the surface beforehand.

Response 15: Thanks a lot for this valuable comment. To response this comment, the sample information has been added to the experimental section in the main text.

"Scanning electron microscopy (SEM), transmission electron microscopy (TEM) images and high-resolution TEM (HR-TEM) images along with element analysis mapping were recorded using Hitachi S-8010 and Titan Themis Cubed G2300 instruments, respectively. The catalysts grown on NF are used as the SEM samples. To prepare the TEM sample, the catalyst was removed from NF by sonication in ethanol solution for 1h and a drop of supernatant was deposited on a duplex Cu mesh. X-ray diffraction (XRD) patterns were measured using a Bruker D8 Advance instrument with a Cu K α radiation source ($\lambda = 1.54178 \text{ \AA}$). The XRD samples were synthesized without the addition of NF in the reactor. X-ray photoelectron spectroscopy (XPS) measurements were conducted with a Thermo Fisher Escalab 250Xi instrument. The catalysts grown on NF (see Supplementary Fig. 1) are used as the XPS samples."

It shall be noted that the catalysts uniformly and densely cover the NF surface, as illustrated in **Supplementary Fig. 1**. So, the influence of NF to XPS results is almost negligible.

Supplementary Fig. 1. SEM images of XPS samples: (a) NiMn-LDH/NF and (b) NiFe-LDH/NF.

Comment 16: Could authors tentatively assign the different peaks of the XRD pattern to the specific crystalline phases?

Response 16: Thanks a lot for this professional comment. To response this comment, we have updated **Supplementary Fig. 3**, in which each diffraction peak is assigned to specific crystalline phase.

Supplementary Fig. 3. XRD patterns of NiMn-LDH and NiFe-LDH.

Comment 17: Line 116: “The smaller Tafel slope indicates faster MOR kinetics on NiMn than on NiFe-LDH”. This is true if they follow the same mechanism and share similar rate-determining steps. However, I wonder if the different MOR mechanisms proposed for these catalysts might affect the Tafel slope obtained experimentally.

Response 17: Thanks a lot for this very professional comment. It is true that the comparison between Tafel slopes shall base on the same mechanism that share similar rate-determining steps. In our case, the MORs on NiMn and NiFe-LDH are found to have similar pathway but different rate-determining steps. So, the comparison between their MOR Tafel slopes is meaningless. To response this comment, we have deleted the Tafel plot in Figure 2b as well as the related discussion in the revised manuscript.

Comment 18: Figure 2C, ECSA: It is not clear in the first instance the presence of two y-axes. I suggest to modify this figure in order to better differentiate the different data.

Response 18: Thanks a lot for this useful comment. To response this comment, we have modified the Figure 2c (now **Figure 2b** in the revised manuscript) to clarify the different data with different y-axes. Note that the *ECSA* and specific activities of NiMn and NiFe-LDH have been updated based on the renewed C_{dl} and C_s . The related discussion has been updated.

Figure 2b. Electrocatalytic performance of the catalysts. (b) ECSA normalized specific activities of NiMn and NiFe-LDHs for MOR (at 1.45 V_{RHE}) and OER (at 1.65 V_{RHE}).

“To rule out the impacts of LDH nanostructures and estimate the intrinsic activities of NiMn and NiFe towards MOR, their current densities are normalized by the electrochemical active surface area (ECSA) (1.22 and 1.17 cm² for NiMn and NiFe-LDH, respectively). Their ECSA are estimated by using an equation of $ECSA = C_{dl}/C_s$, where C_{dl} and C_s are double-layer capacitance (see Supplementary Fig. 7 and note 4) and specific capacitance (see Supplementary Fig. 8), respectively. As plotted in Fig. 2b, NiMn delivers a specific activity of 250.1 mA cm_{ECSA}⁻² towards MOR (at 1.45 V_{RHE}), almost 3.3-fold higher than that (75.8 mA cm_{ECSA}⁻²) of NiFe.”

Comment 19: There is a mention to EWS abbreviation in Figure S9, but it is not defined anywhere. Is this for electrochemical water splitting? If so, this has been defined as overall water splitting (OWS) in the manuscript.

Response 19: Thanks a lot for pointing out this carelessness. Generally, the electrolysis consisting of both HER and OER is denoted as overall water splitting (OWS). Electrochemical water splitting (EWS) usually refers to various electrolysis consisting of HER and OER or alternative electro-oxidations of organics. To avoid confusing, we did not use the term EWS anymore in the revised SI and modified the Figure caption of **Supplementary Fig. 9** (now **Supplementary Fig. 11** in the revised SI) as follows.

“Supplementary Fig. 11. (a) Cell LSV curves of HER/MOR electrolysis using an electrode pair of Pt/C//NiMn-LDH in a single or double-chamber electrolyzer with 1 M KOH and 3 M CH₃OH.”

Comment 20: Could authors include a few brief sentences about the potential of this work to understand further alcohol oxidation reactions with this type of electrocatalysts? How they envision these results are relevant to other reactions beyond methanol?

Response 20: Thanks a lot for this valuable comment. To response this comment, we have added a few brief sentences at the end of **Discussion** section as follows.

“Thus, our work presents a clear understanding on the MOR mechanism of nickel-based hydroxides, which could be expanded to encompass the electro-oxidations of various primary and secondary alcohols that have at least one hydrogen on the carbon attached to the hydroxyl group. The survey on the bifunctional mechanism of MOR provides a new principle for catalyst design in the field of electro-oxidations of alcohols.”

Comment 21: I am not sure “deuterio” is the correct term. I would stick the use to “deuterium”, but I could be mistaken.

Response 21: Thanks a lot for this useful comment. Yes, the term “deuterium” is more popular than “deuterio”. To response this comment, we have used the term “deuterium” to replace “deuterio” in the revised manuscript.

Comment 22: Line 191: “Methanol-induced reduction of Ni^{III}-OOH was suppressed due to the more robust O-D bond”. Could authors further discuss this idea? Do they mean the OD bond in methanol or in the oxyhydroxide? I would also not say “suppressed” as the Raman bands do not seem as sharp and intense as in the OER case, so there might be still some reaction going as the KIE experiments also indicate.

Response 22: Thanks a lot for this constructive comment. In fact, the experimental observations suggest that the *in-situ* formed Ni^{III}-OOH could be reduced to Ni^{III}-(OH)₂ through the hydrogen transfer from methanol to the electrophilic oxygen species of Ni^{III}-OOH. When CH₃OH is replaced by CD₃OD, the reduction of Ni^{III}-OOH is retarded probably because of more robust O-D or C-D bond and hence “slower” deuterium transfer from CD₃OD to Ni^{III}-OOH. Nevertheless, based on only the experimental results, it is hard to determine which bond breaking (i.e., O-D or C-D bond) leads to the reduction of Ni^{III}-OOH. Furthermore, DFT computations can well answer this question (*vide infra*). As shown in the **Supplementary Fig. 24**, the reduction of NiMn(-H) (i.e., NiMn-LDH with Ni^{III}-OOH species) to NiMn (i.e., NiMn-LDH) proceeds through a hydrogen transfer from *OCH₃ to the Ni=O moiety. To response this comment, we have modified the discussion related to the *operando* Raman measurements for the deuterium systems.

In addition, considering the slower reduction kinetics of NiMn(-H) with CD₃OD, we use the word “retard” to replace the word “suppress” for giving more suitable description.

“As shown in Fig. 3a, more pronounced Ni^{III}-O bands (at 473 and 551 cm⁻¹) were detected in the potential range of 1.42~1.52~1.32 V_{RHE} with deuterium media, indicating that the methanol-induced reduction of Ni^{III}-OOH was retarded due to the more robust O-D or C-D bond of the absorbate.”

Comment 23: Do the experiments with deuterium are carried out after some equilibration time or the exchange between H and D in NiOH₂ is instantaneous after immersing into the solution?

Response 23: Thanks a lot for this professional comment. In fact, the catalyst modified anode was immersed into the alkaline CD₃OD (3M)/D₂O (with 1M KOH) and stood by 10 min before measurement. In the liquid phase, the exchange between OH⁻ of KOH and OD⁻ is very fast due to the Grotthuss effect (*Nature* **1999**, 397, 601-604), which was estimated with a time scale of ps or ns (*J. Chem. Phys.* **1995**, 103, 150). The D/H exchange between the catalyst surface and deuterium solution may be not significant based on the TOF-SIMS results (Figure 3B) in the literature (*Chem* **2020**, 6, 2974-2993). As the TOF-SIMS instrument is not available recently, we didn't perform similar measurements. To response this comment, we have added the information of equilibrium time to the **Supplementary note 15** in the revised SI.

“To perform electrochemical measurements with deuterium media, we prepared D₂O solution containing 3 M CD₃OD and 1 M KOH. To reach equilibrium between D₂O and KOH, the anode was immersed in the deuterium media for 10 min before electrochemical measurement.”

Figure 3 (B) in *Chem* **2020**, 6, 2974-2993: Negative TOF-SIMS spectra of O, OH, and OD for β -Ni(OH)₂, β -Ni(OH)₂ soaking in 1 M KOH with 0.5 M CD₃OD, and β -Ni(OH)₂ after NOR in 1 M KOH with 0.5 M CD₃OD.

Comment 24: How was the preparation of the D₂O? Having well-purified D₂O seems to be essential for good electrocatalytic studies such as reported in <https://doi.org/10.1038/s41557-022-01084-y>

Response 24: Thanks a lot for this valuable comment. The D₂O (99.9% atom% D) was purchased from Innochem Co. Ltd. The preparation method is unknown. To response this comment, we have checked the purity of this D₂O by using a NMR method (*Modern Instrs.* **2009**, 1, 23-25). Briefly, a certain amount of secondary deionized water (i.e., 0.04 mL, 0.08 mL, 0.12 mL, 0.16 mL and 0.20 mL) was sequentially added into a 5 mL D₂O solution. After each addition, the mixture of H₂O/D₂O was shaken well and 0.25 mL mixture was picked up as solution A.

Solution B was prepared by adding 6.4 mg 3-(trimethylsilyl)-1-propane sulfonic acid sodium salt (DSS: 97% from Aladdin Co. Ltd.) into 0.5 mL D₂O. To prepare the NMR samples 1-5#, 0.25 mL solution A and 0.30 mL solution B were mixed and added into a NMR tube. To prepare NMR sample 0#, 0.25 mL D₂O and 0.30 mL solution B were mixed and added into a NMR tube. For all samples, the DSS with the same concentration was used as the internal standard. The ¹H NMR spectra for all samples were recorded under the same conditions (¹H 400 MHz NMR). The ¹H NMR spectrum of sample 0# is provided in the **Figure for response 24** below. For each sample, we calculated the ratio of integrated ¹H of H₂O to that of DSS, which is denoted as “A”. Then, as shown in the **Figure for response 24** below, a plot of A and the concentration (Δc , at.% H) of additionally added H₂O was linearly fitted to yield a function of $A = 2.49\Delta c + 4.98$. When $A = 0$, the absolute value of Δc (2 at.% H) is the actual H content in D₂O. So, the atomic D content of D₂O is 99.8 at.%, very close to the labeled purity (99.9% at.% D) of D₂O.

Figure for response 24: (a) ¹H NMR spectrum of sample 0#, and (b) the plot of A and Δc .

Comment 25: Figure S18 for NiMn in D₂O: the Ni processes show two distinctive peaks, with the second one at about 1.45 V. What is the origin of this process? It looks like it does not appear in the same experiment carried out in H₂O.

Response 25: Thanks a lot for this professional comment. In fact, NiMn-LDH has been proved to have high supercapacitor performance (*Adv. Funct. Mater.* **2020**, *30*, 1908223). So, it displays typical electrochemical features of pseudocapacitive materials. That is, their CV responses are quasi-rectangular and “if peaks are present, they are broad and exhibit a small peak-to-peak voltage separation” (*Energy Environ. Sci.* **2014**, *7*, 1597). As shown in **Supplementary Fig. 19a** and Figure 4b (in *Adv. Funct. Mater.* **2020**, *30*, 1908223), the NiMn-LDH indeed displays a quasi-rectangular CV response with “a small peak-to-peak voltage separation” either at the scan rate of 1 mV s⁻¹ in alkaline aqueous solution or at the scan rate of 5 mV s⁻¹ in alkaline CD₃OD/D₂O. We believe that the quasi-rectangular CV response with “a small peak-to-peak voltage separation” is associated with the slower ion diffusion at lower scan rate or deuterium media. Based on the literatures (*Nat. Mater.* **2008**, *7*, 845-854;

ACS Appl. Mater. Interfaces **2017**, *9*, 8649-8658), either the broad quasi-rectangular CV response or that with “a small peak-to-peak voltage separation” originates from successive multiple surface redox reaction of pseudocapacitive material.

To response this comment, we have added additional note in **Supplementary note 9**.

“Due to the pseudocapacitive property, NiMn-LDH usually shows broad CV response or that with a small peak-to-peak voltage separation.”⁸”

Comparison between **Supplementary Fig. 19a** (in our work) and **Figure 4b** (in *Adv. Funct. Mater.* **2020**, *30*, 1908223).

Supplementary Fig. 19a: CV curves of (a) NiMn in H₂O and D₂O solutions both with 1 M KOH with a scan rate of 10 mV s⁻¹.

Figure 4b (in *Adv. Funct. Mater.* **2020**, *30*, 1908223): Electrochemical performance of the Ov-LDH (NiMn-LDH) electrode. (b) CV curves of Ov-LDH (NiMn-LDH) at scan rates 1, 5, 10, 20, and 50 mV s⁻¹.

Comment 26: Would the authors be able to discuss how the use of D₂O could vary the electrical double layer or the potential of zero charge? Could these phenomena be partially responsible for the differences observed for the Ni(II)/Ni(III) processes between aqueous and deuterium media?

Response 26: Thanks a lot for this valuable comment. In fact, NiMn-LDH shall be a pseudocapacitive material because it involves faradaic charge-transfer reaction under the applied potential (*Energy Environ. Sci.* **2014**, *7*, 1597). Its CV response in the Faradaic range is the superposition of surface Faradaic peaks and capacitive double layer charging (*ACS Appl. Mater. Interfaces* **2017**, *9*, 8649-8658). To response this comment, we also provide the CV profiles measured in deuterium media in the non-Faradaic range (see **Supplementary Fig. 21**) and calculate the corresponding C_{dl} . It is found that the C_{dl} values measured in aqueous media and deuterium media are almost same. So, the use of D₂O may not significantly vary the electrical double layer or the potential of zero charge. We believe that the use of D₂O could vary the faradaic charge-transfer reaction (i.e., Ni(II)/Ni(III) oxidation) on the catalyst surface. We also try to explain the origin of the difference observed for the Ni(II)/Ni(III) oxidation

between aqueous and deuterium media in **Supplementary note 10**.

“In the D₂O solution containing 1 M KOH, OD⁻ is much more dominant than OH⁻ due to the fast H/D exchange. It has been generally believed that the oxidation of nickel hydroxide in alkaline media is limited by the OH⁻ adsorption on the catalyst surface. Therefore, the easier Ni^{II}/Ni^{III} oxidation in the deuterium media might be attributed to the more facile adsorption of OD⁻ than OH⁻ on the catalyst surface due to the more polar nature of OD⁻ than OH⁻, which then leaves as water and yields the oxidation of Ni^{II}-(OH)₂ to Ni^{III}-OOH.”

Supplementary Fig. 21. CV curves of (a,b) NiMn-LDH and (c,d) NiFe-LDH measured in the non-Faradaic range in aqueous and deuterium media, respectively.

Plots of current density (J) and scan rate (v) for (e) NiMn-LDH and (f) NiFe-LDH.

Comment 27: Line 272: “Fe^{III} doping suppresses the Ni(II)/Ni(III) oxidation”. I do not think suppresses is the right word as the process is still happening but at a higher potential.

Response 27: Thanks a lot for this valuable comment. To response this comment, we use the word “retards” to replace “suppress” for describing the oxidation of NiFe-LDH at a higher potential.

Comment 28: Were the LSV curves recorded in deuterium media also compensated by iR drop? Does the use of D₂O changes the uncompensated resistance?

Response 28: Thanks a lot for this professional comment. In fact, we have conducted iR compensation the LSV curves recorded in deuterium media. Based on the EIS measurements, respectively, the internal resistance is increased by 0.7 ohm by using deuterium media to replace aqueous media. For instance, the Nyquist plots obtained from the EIS measurements in CH₃OH (3M)/H₂O and CD₃OD (3M)/D₂O, respectively, are provided in **Supplementary Fig. 35**.

Supplementary Fig. 35. Nyquist plots were obtained from EIS measurements for (a) NiMn and (b) NiFe-LDH (both grown on NF) under the MOR conditions (3 M CH₃OH or CD₃OD and 1 M KOH). Inset shows the equivalent circuit, in which R_s corresponds to the internal resistance of electrolyte solution. The data show that the internal resistance is increased by 0.7 ohm by using deuterium media to replace aqueous media.

Reviewer #3

General Comment: The authors explored a bifunctional mechanism of methanol-to-formate electro-oxidation on NiFe/NiMn-based hydroxides. The results reveal that Mn-doped NiOOH NiMn-LDH has a better catalytic activity than the NiFe-LDH based anodic catalyst. However, I do not recommend this paper can be published in Nature Communications since no insights into NiOOH-based catalytic

mechanism. Specific issues to be addressed are:

Comment 1: No new insights can be found in this work. Mn and Fe-embedded NiOOH-based catalysts have been well reported. Also, why NiMn-LDH has a better catalytic performance than NiFe-LDH? The insight into the catalytic mechanism is essential for the design of the catalyst.

Response 1: It is true that the Mn and Fe-embedded NiOOH-based catalysts towards ethanol electro-oxidation reaction (EOR) have been reported by the literature (*Chem.* **2020**, *6*, 2974-2993). In that literature, wang et al. discussed the dopant effects on the redox transition of $\text{Ni}^{\text{II}}\text{-(OH)}_2/\text{Ni}^{\text{III}}\text{-OOH}$ based on the experimental results only. The authors concluded that Fe/Al doping may require more energy for the dehydrogenation of $\text{Ni}^{\text{II}}\text{-(OH)}_2$, while Co/Mn may require less energy for the same process, as compared to that of pristine nickel hydroxide catalyst (see TOC in *Chem.* **2020**, *6*, 2974-2993). Nevertheless, the study on alcohol oxidation mechanism is absent. In other words, it is not clear how the $\text{Ni}^{\text{III}}\text{-OOH}$ interacts with the alcohol to yield corresponding acid.

TOC in *Chem.* **2020**, *6*, 2974-2993.

In another work (*Adv. Energy Mater.* **2021**, *11*, 2101858), Shalom et al. focused on the MOR process on the catalyst of NiFe oxide and proposed the MOR mechanism by experiments only. The authors concluded that the MOR is bottlenecked by the C-H bond breaking in methanol. Although the authors also proposed the in-situ formed $\text{Ni}^{\text{III}}\text{-OOH}$ as the active species towards MOR, clear identification of active site is missing.

In a more recent work (*Appl. Catal. B: Environ.* **2021**, *283*, 119510), Hao et al. proposed a more complete mechanism for the MOR on nickel hydroxide catalyst based on a combined experimental and computational study. As shown in **Figure 7b**, the electrophilic oxygen of $\text{Ni}^{\text{III}}\text{-OOH}$ was proposed as the single active site for MOR. Such a mechanism can well explain the selective formation of formate but cannot explain the rapid diminishing of $\text{Ni}^{\text{III}}\text{-OOH}$ during MOR, as observed in the experiments. To explain both these phenomena, alternative mechanism is required.

Fig. 7b. Gibbs free energy diagrams for the electrocatalytic MeOH conversion process (in *Appl. Catal. B: Environ.* **2021**, 283, 119510).

In our work, we have proposed a bifunctional mechanism for the MOR on nickel hydroxide based catalyst based on a combined experimental and computational study, which involves two combined active sites in Ni^{III}-OOH and differs from the previously reported mechanism involving only a single active site. More importantly, such a bifunctional mechanism can well account for not only the selective formation of formate but also the rapid diminishing of Ni^{III}-OOH during MOR. Moreover, based on a combined experimental and computational study, we have clarified the different effects of Mn, Fe dopants on the overall MOR, including the redox transitions of Ni^{II}-(OH)₂/Ni^{III}-OOH and the contaminant MOR. A brief summary on doping effects is as follows: (i) The Mn doping leads to lower energy barrier for Ni^{II}-(OH)₂ oxidation to Ni^{III}-OOH, as compared to the Fe doping (see **Fig. 5b** for ΔG_{ox} comparison). (2) The Mn doping results in non-spontaneous MOR, while the Fe doping leads to fully spontaneous MOR (see **Fig. 4** for H/D KIE studies and **Fig. 5c** for ΔG change during the MOR).

To highlight the new insight in our work and clarify the difference between our proposed bifunctional mechanism and literature reported mechanisms, we have modified the Abstract and Introduction section.

Abstract. “For nickel-based catalysts, in-situ formed nickel oxyhydroxide has been generally believed as the origin for anodic biomass electro-oxidations. However, rationally understanding the catalytic mechanism still remains challenging. In this work, we demonstrate that NiMn hydroxide as the anodic catalyst can enable methanol-to-formate electro-oxidation reaction (MOR) with a low cell-potential of 1.33/1.41 V at 10/100 mA cm⁻², a Faradaic efficiency of nearly 100% and a good durability in alkaline media, remarkably outperforming NiFe hydroxide. Based on a combined experimental and computational study, we propose a cyclic pathway that consists of reversible redox transitions of Ni^{II}-(OH)₂/Ni^{III}-OOH and a concomitant MOR. More importantly, it is proved that the Ni^{III}-OOH provides combined active sites including Ni^{III} and nearby electrophilic oxygen species, which work in a cooperative manner to promote either spontaneous or non-spontaneous MOR process. Such a bifunctional mechanism can well account for not only the highly selective formate formation but also the transient presence of Ni^{III}-OOH. The different catalytic

activities of NiMn and NiFe hydroxides can be attributed to their different oxidation behaviors. Thus, our work provides a clear and rational understanding for the overall MOR mechanism on nickel-based hydroxides, which is beneficial for advanced catalyst design.”

Comment 2: The concentration of Fe or Mn significantly affects the overpotential and reaction kinetics?

Response 2: Thanks a lot for this valuable comment. To response this comment, we synthesized a series of $\text{Ni}_{1-y}\text{M}_y\text{-LDH}$, where y represents the relative molar content of $\text{M}^{\text{x+}}$ precursor. Their LSV curves were recorded under the MOR conditions and provided in **Supplementary Fig. 33**. As shown in the **Supplementary Fig. 33a**, the $\text{Ni}_{0.80}\text{Mn}_{0.20}\text{-LDH}$ performs best, followed by $\text{Ni}_{0.75}\text{Mn}_{0.25}$, $\text{Ni}_{0.85}\text{Mn}_{0.15}$, $\text{Ni}_{0.90}\text{Mn}_{0.10}$ and $\text{Ni}_{0.95}\text{Mn}_{0.05}\text{-LDH}$. As for NiFe-LDH (**Supplementary Fig. 33b**), $\text{Ni}_{0.75}\text{Fe}_{0.25}$ and $\text{Ni}_{0.80}\text{Fe}_{0.20}\text{-LDH}$ show very comparable performance, followed by $\text{Ni}_{0.85}\text{Fe}_{0.15}$, $\text{Ni}_{0.90}\text{Fe}_{0.10}$ and $\text{Ni}_{0.95}\text{Fe}_{0.05}\text{-LDH}$. So, in this work, we focused on the best performed $\text{Ni}_{0.80}\text{Mn}_{0.20}$ and $\text{Ni}_{0.80}\text{Fe}_{0.20}\text{-LDH}$, both of which were synthesized with the same precursors ratio. For simplicity, $\text{Ni}_{0.80}\text{Mn}_{0.20}$ and $\text{Ni}_{0.80}\text{Fe}_{0.20}\text{-LDH}$ are denoted as NiMn and NiFe-LDH in the main text. We also provided above information in **Supplementary note 15**.

*“To optimize the M content in NiM-LDH (M = Mn, Fe), we synthesized a series of $\text{Ni}_{1-y}\text{M}_y\text{-LDH}$, where y represents the relative molar content of $\text{M}^{\text{x+}}$ precursor. Their LSV curves were recorded under the MOR conditions (3 M CH_3OH and 1 M KOH). As shown in the **Supplementary Fig. 33a**, the $\text{Ni}_{0.80}\text{Mn}_{0.20}\text{-LDH}$ performs best, followed by $\text{Ni}_{0.75}\text{Mn}_{0.25}$, $\text{Ni}_{0.85}\text{Mn}_{0.15}$, $\text{Ni}_{0.90}\text{Mn}_{0.10}$ and $\text{Ni}_{0.95}\text{Mn}_{0.05}\text{-LDH}$. As for NiFe-LDH (**Supplementary Fig. 33b**), $\text{Ni}_{0.75}\text{Fe}_{0.25}$ and $\text{Ni}_{0.80}\text{Fe}_{0.20}\text{-LDH}$ show very comparable performance, followed by $\text{Ni}_{0.85}\text{Fe}_{0.15}$, $\text{Ni}_{0.90}\text{Fe}_{0.10}$ and $\text{Ni}_{0.95}\text{Fe}_{0.05}\text{-LDH}$. So, in this work, we focused on the best performed $\text{Ni}_{0.80}\text{Mn}_{0.20}$ and $\text{Ni}_{0.80}\text{Fe}_{0.20}\text{-LDH}$, both of which were synthesized with the same precursors ratio. For simplicity, $\text{Ni}_{0.80}\text{Mn}_{0.20}$ and $\text{Ni}_{0.80}\text{Fe}_{0.20}\text{-LDH}$ are denoted as NiMn and NiFe-LDH in the main text.”*

Supplementary Fig. 33. LSV curves (with iR compensation) of a series of (a,b) $\text{Ni}_{1-y}\text{Mn}_y\text{-LDH}$ and (c,d) $\text{Ni}_{1-y}\text{Fe}_y\text{-LDH}$ under the MOR conditions (3 M CH_3OH and 1

M KOH). Ni(OH)₂ nano-catalyst synthesized using the same method is used as reference.

Comment 3: Since the active site is Ni, how embedded Mn or Fe contributes to the electrochemical reaction differently?

Response 3: Thanks a lot for this valuable comment. In fact, based on the computational results, the *in-situ* formed Ni^{III}-OOH provides combined active sites towards MOR, including the Ni^{III} and nearby electrophilic oxygen species. As revealed by the combined electrochemistry and computational study, the Mn doping facilitates the redox transition from Ni^{II}-(OH)₂ to Ni^{III}-OOH, while Fe doping retards such a redox transition. Thus, as compared to the NiFe-LDH, the formation of Ni^{III}-OOH on the surface of NiMn-LDH occurs at a lower potential, triggering the MOR at a lower potential.

To probe how the Mn and Fe dopants differently contributes to the redox transition of Ni^{II}-(OH)₂ to Ni^{III}-OOH, we have calculated the partial density of states (PDOS) of Ni²⁺ in NiMn and NiFe-LDH. As shown in the **Supplementary Fig. 23**, NiMn-LDH delivers a *d*-band center at -2.75 eV, more approaching to the Fermi level as compared to that (-2.98 eV) of NiFe-LDH. This result indicates that the Mn doping may more efficiently enhance the OH⁻ adsorption on the Ni site in alkaline media, which leaves as H₂O and yields the Ni^{III}-OOH species. It has been generally believed that the Ni^{III}-OOH formation is limited by the OH⁻ adsorption on Ni site (*Angew. Chem. Int. Ed.* **2021**, *60*, 3095 – 3103). So, the Mn and Fe dopants differently contribute to the Ni^{III}-OOH formation due to their different effects on tuning the *d*-band center of Ni site.

Supplementary Fig. 23 Projected density of states (PDOS) of Ni-3*d* electrons calculated for NiMn and NiFe-LDH.

Related discussion has been added to the main text.

“To understand the different effects of Mn and Fe dopants on the Ni^{II}/Ni^{III} oxidation,

the projected density of state (PDOS) of Ni-3d electrons was calculated for both LDHs. As shown in Supplementary Fig. 23, the d-band center (-2.75 eV) of NiMn-LDH is more approaching to the Fermi level as compared to that (-2.98 eV) of NiFe-LDH, indicating the stronger adsorption of OH⁻ on the surface of NiMn-LDH and hence verifying the lower ΔG_{Ox} relative to that of NiFe-LDH.”

Comment 4: The Gibbs free energy should be converged to 4.92 eV (4x1.23) shown in Figures S26 and S27? This needs to be clarified.

Response 4: Thanks a lot for this professional comment. The standard equilibrium potential of OER in acidic condition is 1.23 V (vs. SHE) and it becomes (1.23-0.0592*pH) V for different pH values according to the Nernst equation. Thus, the equilibrium potential is 0.401 V vs. SHE for pH = 14. Accordingly, the ΔG value was previously calculated by converging to 4*0.401 eV = 1.604 eV. On the other hand, the standard equilibrium potential remains 1.23 V vs. RHE regardless of the conditions. Thus, we provide both the Gibbs free energy profiles with (1.604 eV) and without (4.92 eV) the pH correction in the revised SI for comparison, as shown in the new **Supplementary Fig. 29, 30**. It is worth noting that the pH correction of the reaction free energy does not affect the value of overpotential (η). Taking the Fe site of NiFe(-5H) as an example, without pH correction, $\eta = (1.77 - 1.23) \text{ V} = 0.54 \text{ V}$; with pH correction, $\eta = (0.94 - 0.401) \text{ V} = 0.54 \text{ V}$.

Supplementary Fig. 29. Gibbs free energy diagram for OER on the Ni and Fe sites of NiFe(-5H), respectively, (a) with and (b) without pH correction.

Supplementary Fig. 30. Gibbs free energy diagram for OER on Ni and Mn sites of NiMn(-5H) (a) with and (b) without pH correction, respectively.

REVIEWERS' COMMENTS

Reviewer #2 (Remarks to the Author):

The authors have addressed my comments very well and have carried out an efficient and complete revision of the manuscript, which has been significantly improved.

I am happy to support the publication of this manuscript after the authors address the following minor question:

- When converting potentials to the RHE, authors do not include any pH correction term and consider the RHE and SHE potentials to be equal. Considering that the solution is aqueous and strongly alkaline, is there any reason to do this? I have always seen a $0.059 \cdot \text{pH}$ (V) term included.

Reviewer #3 (Remarks to the Author):

The authors answered my questions well, and I would like to recommend that this work be published in Nature Communications. Just one minor comment.

One paper (<https://doi.org/10.1038/s41467-020-18459-9>) recently published in Nature Communications reports MOR mechanism based on NiOOH NRs. The authors stated, "Nevertheless, the study on alcohol oxidation mechanism is absent" in the rebuttal letter. A brief comparison between the current and reported works should be advanced for publication in NC.

09th-Mar.-2023

Journal: Nature communications

Manuscript Number: NCOMMS-22-36007B

Title: Unraveling a bifunctional mechanism for methanol-to-formate electro-oxidation on nickel-based hydroxides

Dear Editor,

Sincerely thanks for your consideration of, in principle, acceptance and kind response as well as the comments from reviewers 2-3#. We have carefully responded to your and reviewers' comments on the following pages and made further modifications in the manuscript. The changes in the text have been highlighted in yellow color.

We hope that a revised version of the manuscript is suitable for publication in *Nature communications*.

Sincerely yours,

Lai Feng

College of Energy

Soochow University, Suzhou 215006, China

REVIEWERS' COMMENTS

Reviewer #2 (Remarks to the Author):

The authors have addressed my comments very well and have carried out an efficient and complete revision of the manuscript, which has been significantly improved.

I am happy to support the publication of this manuscript after the authors address the following minor question:

- When converting potentials to the RHE, authors do not include any pH correction term and consider the RHE and SHE potentials to be equal. Considering that the solution is aqueous and strongly alkaline, is there any reason to do this? I have always seen a $0.059 \cdot \text{pH}$ (V) term included.

Reviewer #3 (Remarks to the Author):

The authors answered my questions well, and I would like to recommend that this work be published in Nature Communications. Just one minor comment.

One paper (<https://doi.org/10.1038/s41467-020-18459-9>) recently published in Nature Communications reports MOR mechanism based on NiOOH NRs. The authors stated, "Nevertheless, the study on alcohol oxidation mechanism is absent" in the rebuttal letter. A brief comparison between the current and reported works should be advanced for publication in NC.

REVIEWER COMMENTS

Reviewer #2 (Remarks to the Author):

The authors have addressed my comments very well and have carried out an efficient and complete revision of the manuscript, which has been significantly improved.

I am happy to support the publication of this manuscript after the authors address the following minor question:

- When converting potentials to the RHE, authors do not include any pH correction term and consider the RHE and SHE potentials to be equal. Considering that the solution is aqueous and strongly alkaline, is there any reason to do this? I have always seen a $0.059 \times \text{pH}$ (V) term included.

Response: Thanks a lot for this professional comment. Indeed, there is a problem in directly converting the potential value versus SHE to that versus RHE. To response this comment, we provide the calculations for the standard equilibrium potential of MOR versus SHE and that versus RHE under non-alkaline and alkaline conditions as follows:

The methanol to formate electro-oxidation (MOR) delivers 4 electrons per methanol molecule as shown in Eq. (1)

The corresponding Gibbs free energy (ΔG_1^0) of (1) is

$$\Delta G_1^0 = G_{\text{HCOOH}}^0 - G_{\text{CH}_3\text{OH}}^0 + 4 \times G_{\text{H}_2\text{O}}^0 = 40.1 \text{ kJ mol}^{-1}$$

Therefore, according to Faraday's law ($\Delta G^0 = n \times F \times E_{\text{cell}}^0$, where n is the number of transferred electrons, F is the Faraday's constant), the equilibrium potential vs SHE (E_{cell}^0) can be calculated:

$$E_{\text{cell}}^0 = \Delta G_1^0 / (nF) = 40.1 / (4 \times 96,485) = 0.103 \text{ V}_{\text{SHE}}$$

This is in line with literature report (Angew. Chem. Int. Ed. 2021, 60, 3148 – 3155).

However, alkaline MOR follows Eq. (2):

Therefore, the ΔG_2^0 is as follows.

$$\Delta G_2^0 = (3 \times G_{\text{H}_2\text{O}}^0 + G_{\text{HCOOH}}^0) - (G_{\text{CH}_3\text{OH}}^0 + 4 \times G_{\text{OH}^-}^0) = -279.5 \text{ kJ mol}^{-1}$$

According to Faraday's law ($\Delta G^0 = n \times F \times E_{\text{cell}}^0$, where n is the number of transferred electrons, F is the Faraday's constant), the equilibrium potential vs SHE (E_1^0) can be calculated:

$$E_{\text{cell}}^0 = \Delta G_2^0 / (nF) = -279.5 / (4 \times 96,485) = -0.724 \text{ V}_{\text{SHE}}$$

Then, the potential shall be corrected by $\text{pH} = 14$ (1 M KOH) according to the Nernst equation.

$$E_{(2)} = E_{\text{cell}}^0 - 1/4 \times 0.059 \times \log[1/(\text{OH}^-)^4] = E_{\text{cell}}^0 + 0.059 \times \text{pH} - 0.059 \times 14 = E_{\text{cell}}^0 +$$

$0.059 \times \text{pH} - 0.826$, where $E^0_{\text{cell}} = -0.724 \text{ V}_{\text{SHE}}$

For reversible hydrogen electrode (RHE), the internal reaction is as follows:

$$E_{(3)} = E^0 - 1 \times 0.059 \times \text{pH} = -0.059 \times \text{pH}, \text{ where } E^0 = 0.00 \text{ V}_{\text{SHE}}$$

Therefore, the standard equilibrium potential versus RHE of the MOR under alkaline conditions ($\text{pH} = 14$) can be calculated as follows:

$$E_{(\text{vs RHE})} = E_{(2)} - E_{(3)} = [E^0_{\text{cell}} + 0.059 \times \text{pH} - 0.826] - [-0.059 \times \text{pH}] = 0.102 \text{ V}_{\text{RHE}}$$

This standard equilibrium potential ($E_{(\text{vs RHE})} = 0.102 \text{ V}_{\text{RHE}}$), which is numerically close to the equilibrium potential vs SHE ($E^0_{\text{cell}} = 0.103 \text{ V}_{\text{SHE}}$) but not the same, has been used for calculating the overpotential for MOR under alkaline conditions.

Table 1. Summary of Gibb's free energy of formation ($\Delta_f G_m^0$) values (All values are reported under standard conditions of 100 kPa and 298 K).^a

Molecular formula	H ⁺	H ₂ O	OH ⁻	CH ₃ OH	HCOOH
$\Delta_f G_m^0$ (KJ mol ⁻¹)	0.0	-237.1	-157.2	-175.3	-372.3

^a Lide, D. in *CRC handbook of chemistry and physics* (CRC press, 2005).

In addition, in this manuscript, all potentials have been corrected by including the term $0.059 \times \text{pH}$ using the following equation: $E_{(\text{RHE})} = E_{(\text{Ag}/\text{AgCl})} + 0.197 + 0.059 \times \text{pH}$, as described in **Supplementary note 16**.

Reviewer #3 (Remarks to the Author):

The authors answered my questions well, and I would like to recommend that this work be published in Nature Communications. Just one minor comment.

One paper (<https://doi.org/10.1038/s41467-020-18459-9>) recently published in Nature Communications reports MOR mechanism based on NiOOH NRs. The authors stated, "Nevertheless, the study on alcohol oxidation mechanism is absent" in the rebuttal letter. A brief comparison between the current and reported works should be advanced for publication in NC.

Response: We are very thankful for this suggestion. As this literature reported the mechanism for methanol oxidation to CO₂ rather than formate, we did not find it. To response this suggestion, we have modified the statement in the previous rebuttal letter.

"Nevertheless, although the mechanism for methanol-to-CO₂ oxidation has been well studied, the mechanism for methanol-to-formate oxidation is less-explored."

In addition, we also briefly compare the two oxidation mechanisms in computational section as follows.

"This is very different from that reported for the highly selective methanol-to-CO₂ oxidation over NR-Ni(OH)₂, where the dehydrogenation of *CH₂O to *CHO is thermodynamically less favored than that of *COOH to *CO₂.⁴³"

Reference 43

43. Wang X, et al. Materializing efficient methanol oxidation via electron delocalization in nickel hydroxide nanoribbon. *Nat. Commun.* **11**, 4647 (2020).